# Single-cell analyses reveal SARS-CoV-2 interference with intrinsic immune response in the human gut

Sergio Triana[1,2] (iD), Camila Metz-Zumaran[3] (iD), Carlos Ramirez[4], Carmon Kee[3,5], Patricio Doldan[3,5], Mohammed Shahraz[1], Daniel Schraivogel[6], Andreas R Gschwind[7], Ashwini K Sharma[3,4] (iD), Lars M Steinmetz[6,7,8], Carl Herrmann[4] (iD), Theodore Alexandrov[1,9,10,*] (iD), Steeve Boulant[3,5,**] & Megan L Stanifer[11,***] (iD)

## Abstract

Exacerbated pro-inflammatory immune response contributes to COVID-19 pathology. However, despite the mounting evidence about SARS-CoV-2 infecting the human gut, little is known about the antiviral programs triggered in this organ. To address this gap, we performed single-cell transcriptomics of SARS-CoV-2-infected intestinal organoids. We identified a subpopulation of enterocytes as the prime target of SARS-CoV-2 and, interestingly, found the lack of positive correlation between susceptibility to infection and the expression of *ACE2*. Infected cells activated strong pro-inflammatory programs and produced interferon, while expression of interferon-stimulated genes was limited to bystander cells due to SARS-CoV-2 suppressing the autocrine action of interferon. These findings reveal that SARS-CoV-2 curtails the immune response and highlights the gut as a pro-inflammatory reservoir that should be considered to fully understand SARS-CoV-2 pathogenesis.

**Keywords** human intestinal epithelial cells; interferon; intrinsic immune response; SARS-CoV-2; single-cell RNA sequencing
**Subject Categories** Chromatin, Transcription & Genomics; Immunology; Microbiology, Virology & Host Pathogen Interaction
**Mol Syst Biol. (2021) 17: e10232**

## Introduction

Coronavirus disease 2019 (COVID-19) is caused by the severe acute respiratory syndrome coronavirus 2 (SARS-CoV-2). This highly infectious zoonotic virus has caused a global pandemic with more than 105,000,000 people infected worldwide as of February 2021. An exacerbated pro-inflammatory immune response generated by the host has been proposed to be responsible for the symptoms observed in patients (Giamarellos-Bourboulis *et al*, 2020; Mehta *et al*, 2020; Zhou *et al*, 2020c). Numerous studies have correlated the nature and extent of the immune response with the severity of the disease (Chen *et al*, 2020; Lucas *et al*, 2020; preprint: Mathew *et al*, 2020). While many countries have succeeded in curtailing the first wave of infection, the second wave has hit many countries harder than the first. Therefore, it is very urgent that we understand the virus-induced pathogenesis, in particular the immune response generated by the host, to develop prophylactic therapeutics, antiviral approaches, and pharmacological strategies to control and revert the pathologies seen in patients.

SARS-CoV-2 is a member of the betacoronavirus genus, which initiates its lifecycle by exploiting the cellular receptor angiotensin-converting enzyme 2 (ACE2) to enter and infect host cells (Hoffmann *et al*, 2020). Virus entry relies not only on ACE2, but also on the cellular proteases furin and the transmembrane serine protease 2 (TMPRSS*2*) that cleave and activate the SARS-CoV-2 envelope spike protein (Bestle *et al*, 2020). Following release of the genome into the cytosol, translation of the positive-strand RNA genome is initiated and viral proteins quickly induce the formation of cellular membrane-derived compartments for virus replication

1   Structural and Computational Biology Unit, European Molecular Biology Laboratory, Heidelberg, Germany
2   Faculty of Biosciences, Collaboration for Joint PhD Degree between EMBL and Heidelberg University, Heidelberg, Germany
3   Department of Infectious Diseases, Virology, Heidelberg University Hospital, Heidelberg, Germany
4   Health Data Science Unit, Medical Faculty University Heidelberg and BioQuant, Heidelberg, Germany
5   Research Group "Cellular Polarity and Viral Infection", German Cancer Research Center (DKFZ), Heidelberg, Germany
6   Genome Biology Unit, European Molecular Biology Laboratory, Heidelberg, Germany
7   Department of Genetics, Stanford University School of Medicine, Stanford, CA, USA
8   Stanford Genome Technology Center, Palo Alto, CA, USA
9   Skaggs School of Pharmacy and Pharmaceutical Sciences, University of California San Diego, La Jolla, CA, USA
10  Molecular Medicine Partnership Unit (MMPU), European Molecular Biology Laboratory, Heidelberg, Germany
11  Department of Infectious Diseases, Molecular Virology, Heidelberg University Hospital, Heidelberg, Germany
    *Corresponding author. Tel: +49 0 6221 387 8690; E-mail: theodore.alexandrov@embl.de
    **Corresponding author. Tel: +49 0 6221 56 7865; E-mail: steeve.boulant@med.uni-heidelberg.de
    ***Corresponding author. Tel: +49 0 6221 56 7858; E-mail: m.stanifer@dkfz.de

and *de novo* assembly of virus particles (Cortese *et al*, 2020; Klein *et al*, 2020). The host cells execute several strategies to counteract viral replication. Cellular pathogen recognition receptors (PRRs) sense viral molecular signatures or pathogen-associated molecular patterns (PAMPs) and induce a signaling cascade leading to the induction of interferons (IFNs) and pro-inflammatory molecules. IFNs represent the first line of defense against viral infection as their autocrine and paracrine signaling leads to the production of hundreds of interferon-stimulated genes (ISGs) known to exert broad antiviral functions (Stanifer *et al*, 2019, 2020a).

SARS-CoV-2 infection is not limited to the respiratory tract and COVID-19 patients show systemic manifestation of the disease in multiple organs (Gupta *et al*, 2020; Prasad & Prasad, 2020). For many of these organs, it is unclear whether the pathology is a side effect of SARS-CoV-2 infection in the lung and its associated pro-inflammatory response or whether it is due to a direct SARS-CoV-2 infection of the specific organ. For the gastrointestinal (GI) tract, there is clear evidence of SARS-CoV-2 replication which is associated with the release of viral genome into the feces (Wölfel *et al*, 2020; Wu *et al*, 2020; Xiao *et al*, 2020; Xing *et al*, 2020). Human intestinal organoids have been established as a robust model to study SARS-CoV-2 infection and provided direct evidence about primary human intestinal epithelial cells efficiently supporting SARS-CoV-2 replication (Lamers *et al*, 2020; Stanifer *et al*, 2020b; Zang *et al*, 2020). Importantly, while SARS-CoV-2 infection in the lung is characterized by a curtailed IFN response (Blanco-Melo *et al*, 2020; Hadjadj *et al*, 2020), the intrinsic immune response in intestinal epithelial cells is characterized by the production of IFN and ISGs, with IFNs providing some protection to intestinal epithelium cells against SARS-CoV-2 (Stanifer *et al*, 2020b). Studies in human intestinal organoids revealed that only discrete cells are susceptible to SARS-CoV-2 infection, and some evidence suggests that these cells may be enterocytes (Lamers *et al*, 2020). However, the precise cell tropism of SARS-CoV-2 within the colon and other parts of the gastrointestinal tract is yet to be fully characterized. Finally, despite the driving role of inflammation in the pathologies observed in COVID-19 patients, we are still lacking important molecular details concerning the inflammatory response generated by SARS-CoV-2-infected cells and how the surrounding bystander cells will respond to it.

Here, we aim to address the outlined gaps by applying single-cell RNA sequencing to human ileum- and colon-derived organoids infected with SARS-CoV-2. Using differential gene expression analysis and multiplex single-molecule RNA fluorescence *in situ* hybridization (FISH), we investigated the cell type tropism of SARS-CoV-2 and its link to *ACE2* expression levels. While we could show that a subpopulation of enterocytes represents the primary site of SARS-CoV-2 infection, we did not observe correlation between infectivity and *ACE2* expression. Interestingly, we could observe that SARS-CoV-2 infection is associated with a downregulation of *ACE2* expression. Pathway analysis revealed that infected cells mount a strong pro-inflammatory response characterized by the upregulation of both NFκB/TNF expression and the activity of their respective pathways. On the contrary, bystander cells were characterized by an upregulation of the IFN-mediated immune response as monitored by the increased production of ISGs. Importantly, using a combination of multiplex single-molecule RNA FISH and IFN-reporter bioassays we could show that while IFN could act in a paracrine manner in bystander cells, IFN cannot act in an autocrine manner in SARS-CoV-2-infected cells. Our findings demonstrate that SARS-CoV-2 has developed strategies to impair IFN-mediated signaling in infected cells, and together with our previous observations showing that IFN restricts SARS-CoV-2 replication in intestinal cells (Stanifer *et al*, 2020b), these results suggest that SARS-CoV-2 manipulates the cell-intrinsic innate immune response to promote its replication and spread.

# Results

## Single-cell RNA sequencing of SARS-CoV-2-infected colon and ileum organoids

A fraction of COVID-19 patients show enteric symptoms and it has been shown that SARS-CoV-2 replicates in the intestinal tract of patients (Xiao *et al*, 2020) and in human primary intestinal epithelial cells (Lamers *et al*, 2020; Zhou *et al*, 2020a; Zang *et al*, 2020; Stanifer *et al*, 2020b). To characterize SARS-CoV-2 interactions with primary human intestinal epithelial cells (hIECs) human intestinal organoids were infected by SARS-CoV-2. To address whether organoids derived from distinct parts of the intestinal tract display different susceptibility, colon- and ileum-derived organoids were seeded in two dimensions (2D, flat as monolayers on culture dishes,) to access the apical side of the cells as it was previously shown that SARS-CoV-2 infection is mainly through the apical side due to the polarized distribution of ACE2 (Zang *et al*, 2020). To control that our organoids seeded in 2D were properly differentiated, we followed the expression of various stem cells and differentiated cell markers over time following removal of WNT and decreasing the concentrations of R-Spondin and Noggin (Fig EV1A). We found that organoids could properly differentiate in 2D as characterized by the loss of stem cell marker and the increased expression of several markers of differentiated cells which is consistent with previous work (Ettayebi *et al*, 2016; Kolawole *et al*, 2019; Ding *et al*, 2020; Stanifer *et al*, 2020b, 2020c; Zang *et al*, 2020). Differentiated 2D organoids were infected by SARS-CoV-2, and their response was monitored along the course of infection. Through direct visualization of infected organoids, we observed that starting at 36 h post-infection (hpi) cells started to die and that at 48 hpi most cells were dead (Fig EV1B). As a consequence, we limited our analysis of infection to 24 h. Of note, we observed that only a small fraction of the cells were infected with SARS-CoV-2 even when using a higher amount of virus for infection, suggesting that only a discrete population in organoids are permissive to infection. For both colon- and ileum-derived organoids, we could observe the presence of infected cells as early as 4 h post-infection (hpi) with the number of infected cells increasing within the course of infection (Fig EV1C and D). This was corroborated with an increase in intracellular viral RNA and the release of *de novo* infectious virus particles over time (Fig EV1E and F), thus demonstrating efficient virus replication and spreading of infection in both colon and ileum organoids. To characterize how hIECs respond to SARS-CoV-2 infection, we monitored the production of type I IFN (*IFNβ1*) and type III IFNs (*IFNλ1* and *IFNλ2-3*) over time. SARS-CoV-2 did not induce significant production of *IFNβ1* in either ileum or colon organoids, except for a slight upregulation of *IFNβ1* expression in colon organoids at 24 hpi

(Fig EV1G). On the contrary, for *IFNλ2-3*, a strong upregulation was observed in both colon and ileum organoids upon infection by SARS-CoV-2 (Fig EV1I). Interestingly, similar to *IFNβ1*, *IFNλ1* was not upregulated in response to infection (Fig EV1H). Taken together, these results show that a fraction (around 7–10% 24 hpi, Fig EV1D) of human intestinal epithelial cells supports SARS-CoV-2 infection, replication, *de novo* production of infectious virus particles and that infection is associated with the upregulation of type III IFNs (*IFNλ2-3*).

**Determination of SARS-CoV-2 intestinal epithelial cell tropism**

Human intestinal organoids are composed of multiple cell types partially recapitulating the cellular complexity of the human intestinal epithelium. Although it is clear that SARS-CoV-2 infects the human intestinal epithelium, which specific cell types are infected by the virus, how infection impacts the transcriptional landscape of each individual cell type, and how bystander cells respond to viral infection remains unknown. To characterize SARS-CoV-2 interactions with hIECs at the single-cell transcriptional level, colon- and ileum-derived organoids were infected with SARS-CoV-2 as described above and subjected to single-cell RNA sequencing (scRNAseq) (Fig 1A). Single-cell suspensions were generated and 3' scRNAseq was performed across six biological conditions (mock, 12 hpi, and 24 hpi for both colon and ileum organoids). scRNAseq provided broad transcriptional profiles for around 2,000 cells per condition with 5,000 and 6,000 genes profiled on average per cell for the colon and the ileum, respectively (Appendix Fig S1A–H for colon organoids and Appendix Fig S1I–P for ileum organoids).

To identify the cell types present in our organoids, we used unsupervised clustering and Uniform Manifold Approximation and Projection (UMAP) algorithm for visualization of our scRNAseq data (Fig EV2A and F). The clustering revealed the presence of multiple cell subpopulations. Using both differentially expressed cell type-specific marker genes (Figs 1C and EV2B and G) and markers from single-cell atlases of human intestinal tissues (Smillie *et al*, 2019) and from our annotated scRNAseq data from human ileum biopsies (preprint: Triana *et al*, 2020), we could identify the major cell types represented in these populations (Fig 1B). We identified eight and nine major populations of cells in the colon and ileum organoids, respectively (Fig 1B). Stem cells, transient amplifying (TA) cells, enterocytes, goblet, and enteroendocrine cells were found to be present in intestinal organoids (Fig 1B). Importantly, while the proportion of each cell type in our organoids was similar to the one observed in another report (Fujii *et al*, 2018), we found that this ratio was not fully identical to the one observed in primary tissues (biopsies), highlighting the limitation of organoids although they are the current best primary model to mimic the human gut. Different subpopulations of

enterocytes in ileum and colon (*CLCA4+*, *ALDOB+*, *MUC13+*), were identified, namely, enterocytes 1 (*GUCA2A+*, *FABP6+*, *CA4+*), enterocytes 2 (*MMP7+*, *MUC1+*, *CXCL1+*) as well as immature enterocytes 1 and immature enterocytes 2, the latter likely representing cells not fully differentiated into mature enterocytes. The presence of two distinct populations of enterocytes and their immature-related forms is consistent with previous reports (Smillie *et al*, 2019). Importantly, infection by SARS-CoV-2 did not alter cell clustering (Fig 1B, left UMAP inset panels). To address the possible difference between the mock and infected organoids cell types, we calculated the correlation of class-average transcriptional profiles between cell types of both conditions (Fig EV2D and I). This revealed an overall high correlation within every cell type ($r > 0.99$). Furthermore, SARS-CoV-2 infection had no impact on the proportions of the different cell types present in both the colon and ileum organoids (Fig 1D) thus allowing us to investigate cell tropism.

To increase the sensitivity and dynamic range in detecting the SARS-CoV-2 genome and selected host genes, we made use of additional targeted scRNAseq (Schraivogel *et al*, 2020) performed on the same organoid samples. Targeted scRNAseq is more sensitive to detect genes-of-interest irrespective of their expression level and quantifies gene expression with a higher dynamic range compared with conventional 3' scRNAseq (Schraivogel *et al*, 2020). We selected 12 genes, including the SARS-CoV-2 genome, the SARS-CoV-2 receptor *ACE2*, an interferon-stimulated gene (*ISG15*), and a panel of hIEC type markers that we previously determined by scRNAseq of ileum biopsies and organoids (*APOA4*, *CHGB*, *FABP6*, *FCGBP*, *LYZ*, *MKI67*, *OLFM4*, *SLC2A2*, *SMOC2*, and *SST*) (preprint: Triana *et al*, 2020). Looking at the relative expression of SARS-CoV-2 genome in mock vs infected cells (Fig EV3A and B), classical 3' scRNAseq detected the viral genome in a proportion of organoid cells (Fig EV3A (left panel); ~25% at 12 hpi and ~75% at 24 hpi), while using the targeted approach, SARS-CoV-2 counts were detected in virtually all cells in the infected samples (Fig EV3B). Concurrently, we observed that the number of SARS-CoV-2 counts per cell increased over the course of infection in both targeted and whole transcriptome scRNAseq (Fig EV3A and B, right panels), consistent with active replication of the virus in organoids monitored using q-RT–PCR (Fig EV1E). Since immunofluorescence staining was performed in parallel to the scRNAseq samples and revealed that less than 10% of cells were infected (Fig EV1C and D) (Lamers *et al*, 2020; Stanifer *et al*, 2020b), the presence of SARS-CoV-2 genome in all cells likely does not reflect active viral replication, but could be explained by the presence of viruses attached to the cell's surface or by free-floating viral particles or RNA. This is also confirmed by the presence of viral transcripts in the empty droplets (Fig EV3C and D). Capitalizing on the high dynamic range provided by targeted scRNAseq, we defined cells having productive infection and replication as those with the

**Figure 1. Single-cell sequencing of SARS-CoV-2-infected colon- and ileum-derived human organoids.**

A  Schematic representation of the experimental workflow.

B  Uniform manifold approximation and projection (UMAP) embedding of single-cell RNA-Seq data from mock and SARS-CoV-2-infected colon-derived (left panels) and ileum-derived (right panels) organoids colored according to the cell type. Small insets highlight in the UMAP the cell from mock and infected organoids at 12 and 24 hpi (Red) and the rest of the cells (gray).

C  Dot plot of the top marker genes for each cell type for (left) colon- and (right) ileum-derived organoids. The dot size represents the percentage of cells expressing the gene; the color represents the average relative expression across the cell type.

D  Bar plot displaying the proportion of each cell type in mock and infected organoids (12 and 24 hpi).

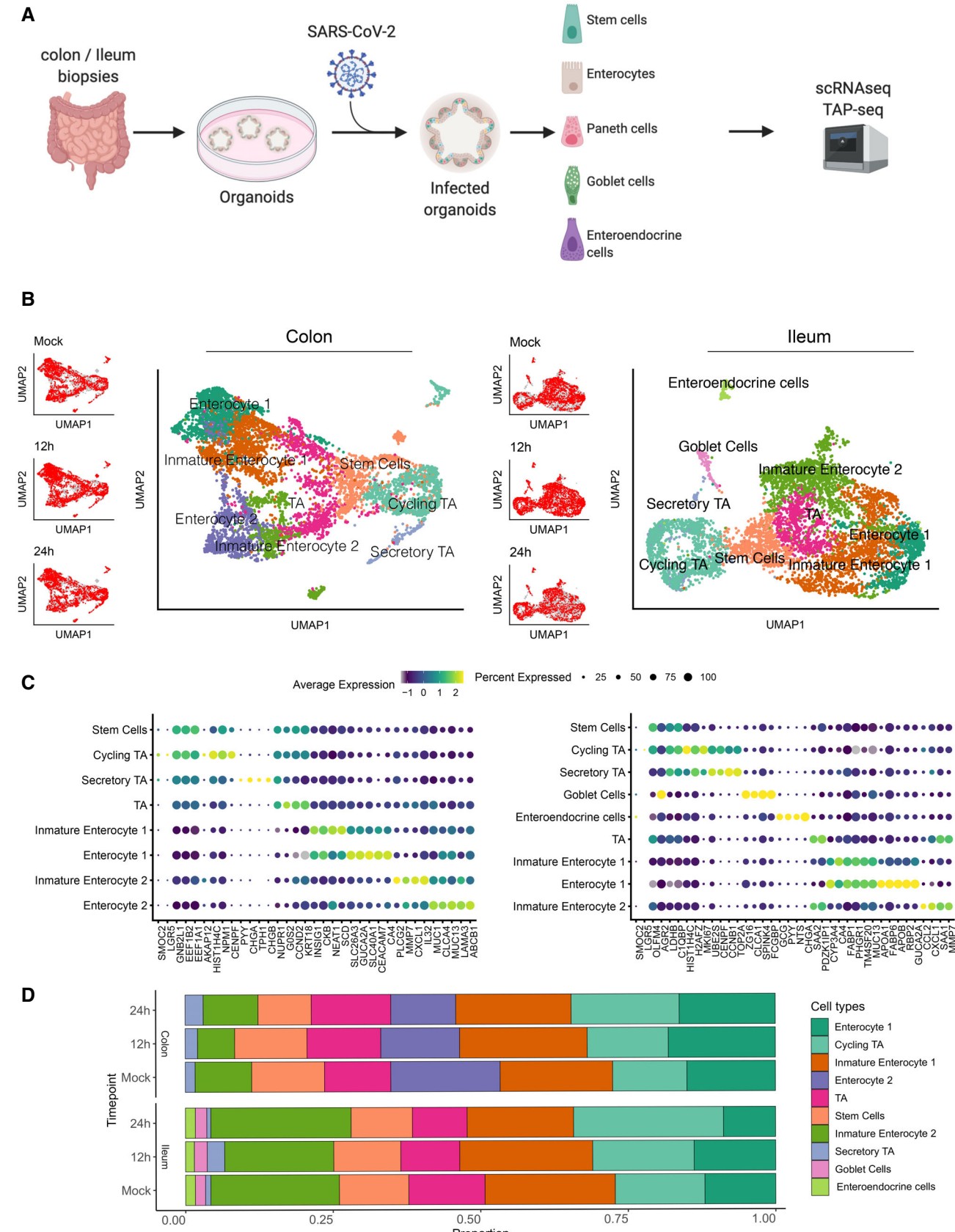

Figure 1.

SARS-CoV-2 counts higher than a baseline, calculated as the mean expression of SARS-CoV-2 measured by targeted scRNAseq in all cell-containing droplets (Fig EV3C and D, top panels). Cells with targeted scRNAseq SARS-CoV-2 expression levels below the estimated base-line level were defined as non-infected bystander cells. Following this approach, we could correct for the presence of contaminating viral RNA. Only a small fraction of cells (Colon: 4.5% and Ileum: 5.3% at 24 hpi) was defined as supporting SARS-CoV-2 infection (Fig EV3C and D) which is consistent with our immunofluorescence staining of SARS-CoV-2-infected organoids (Fig EV1C and D) and other reports (Lamers *et al*, 2020; Stanifer *et al*, 2020b). Additionally, we evaluated SoupX (Young & Behjati, 2020) as an alternative method to threshold SARS-CoV-2 genome amounts and be able to determine infected vs bystander cells. However, using this approach we found that small cells became over-enriched for viral counts as SoupX uses the total count of transcripts per cell for the removal of their predicted contam-ination fraction (Appendix Fig S2). We therefore opted for our targeted scRNAseq and thresholding approaches based on the mean expression of SARS-CoV-2 as this does not bias our genome counts based on cell size and, most importantly, correlates with our quan-tification of number of infected cells using immunofluorescence.

Comparing the targeted scRNAseq and the conventional 3' scRNAseq approaches, we observed high correlation between the expression levels estimated by these technologies (Fig EV3E and F top panels), with targeted scRNAseq providing several orders of magnitude higher dynamic range, compared with the conventional 3' scRNAseq (Fig EV3E and F bottom panels). The high positive correlation between the two sequencing approaches (except for SLCA2) serves as a quality control confirming that targeted scRNAseq could robustly quantify both viral and cellular genes.

While SARS-CoV-2 could be detected in most cell types in both colon and ileum organoids (Fig 2A), immature enterocytes 2 consis-tently represented the main virus-targeted cell type (Fig 2B). The proportion of infected immature enterocytes 2 and the number of SARS-CoV-2 genome copy per cell increased over the course of infection (Fig 2B) which is consistent with both the increasing number of infected cells observed by immunofluorescence and the increasing copies of SARS-CoV-2 genome over time (Figs EV1C–E, and EV3A and B). In ileum-derived organoids, TA cells in the secre-tory lineage were also found to be highly infected by SARS-CoV-2 (Fig 2B, right panel). Interestingly, these cells were infected mostly at 24 hpi, suggesting that they are secondary targets of infection. Taken together, these results suggest that immature enterocytes 2 are the primary target of SARS-CoV-2 infection in hIECs both in colon and ileum.

## SARS-CoV-2 cell tropism and association with expression of ACE2 and TMPRSS2

The angiotensin-converting enzyme 2 (ACE2) and the cellular protease type II transmembrane serine protease 2 (TMPRSS2) are known to be major determinants for SARS-CoV-2 infection. ACE2 is the cellular receptor of SARS-CoV-2-mediating viral entry (Hoff-mann *et al*, 2020). TMPRSS2 is a cellular protease that processes the SARS-CoV-2 spike (S) protein which is an essential step for viral envelope fusion with the host membrane and release of viral contents in the cytosol of the cells. Combined conventional and targeted scRNAseq enabled us to investigate the link between SARS-

CoV-2 genome copy numbers and expression of *ACE2* in a cell type-specific manner. Different from what we have expected, immature enterocytes 2, the main site of SARS-CoV-2 infection in both colon and ileum organoids (Fig 2A and B), were not the cells displaying the highest levels of *ACE2* (Fig EV4A and B). Analysis of *ACE2* expression levels in all cell types revealed that cells with relatively high levels of *ACE2* (*e.g.*, enterocytes 1) were not susceptible to SARS-CoV-2 infection (Fig 2B and C). Similarly, we found that SARS-CoV-2 infection is not associated with the expression of the receptor structural homologue *ACE*, a candidate receptor for SARS-CoV-2 basigin (*BSG*, also known as CD147), as well as the cellular proteases furin, cathepsin L1 (*CTSL*), aminopeptidase *ANPEP* and *DPP4* (MERS-CoV receptors) (Fig 2C). On the contrary, *TMPRSS2* was found to be highly expressed in immature enterocytes 2 (Fig 2C). In summary, although *ACE2* is a recognized receptor for SARS-CoV-2, we found no association between high *ACE2* expres-sion levels and increased susceptibility to infection on the single-cell level or across detected types of hIECs.

## SARS-CoV-2 infection induces downregulation of *ACE2* expression in intestinal organoids

*ACE2* has been reported to act as an interferon-stimulated gene (ISG) resulting in an increased expression level upon viral infection and interferon stimulation of nasal and lung epithelial cells (Ziegler *et al*, 2020). Similarly, in COVID-19 patients, ACE2 expression was shown to be upregulated in lung epithelial cells compared with control patients (Chua *et al*, 2020). To investigate whether the expression of either ACE2 or other putative receptors and key cellu-lar proteases is upregulated upon SARS-CoV-2 infection or upon SARS-CoV-2-mediated immune response in primary hIECs, we compared their expression levels in mock-infected cells *vs*. both SARS-CoV-2-infected and non-infected bystander cells. Although we did not observe any association between *ACE2* expression in non-infected cells and their propensity to be infected by SARS-CoV-2 (Figs 2B and C, and EV4A and B), differential gene expression anal-ysis revealed that upon SARS-CoV-2 infection the *ACE2* expression levels were downregulated (Fig 3A and B). In colon organoids, visi-ble downregulation of *ACE2* expression was observed in infected cells, progressing from 12 hpi to 24 hpi, as compared to mock-infected cells (Fig 3B, left panel). Importantly, no significant dif-ference of *ACE2* expression in the bystander cells was observed (Fig 3B). In ileum-derived organoids, *ACE2* expression was also downregulated in the infected cells. However, in contrast to colon organoids, *ACE2* expression in bystander cells of ileum organoids was also downregulated as compared to mock-infected cells (Fig 3B, right panels). Importantly, ACE2 expression was found to be nega-tively correlated with the presence of the viral genome (Fig 3C and D). The downregulation of *ACE2* expression was not only observed in immature enterocytes 2 which were identified as the primary site of SARS-CoV-2 infection (Fig 2B), but it was also observed in most other cell types present in ileum-derived organoids over the course of SARS-CoV-2 infection (Fig 3E). This is unlikely to be a product of global transcriptional repression as the gene detection rate does not change between conditions (Fig EV4C). Expression levels of the other SARS-CoV-2 putative receptors and of key cellular proteases (e.g. *TMPRSS2*, *furin*, and *CTSL*) were also found to be reduced in both infected and bystander cells in ileum-derived organoids as

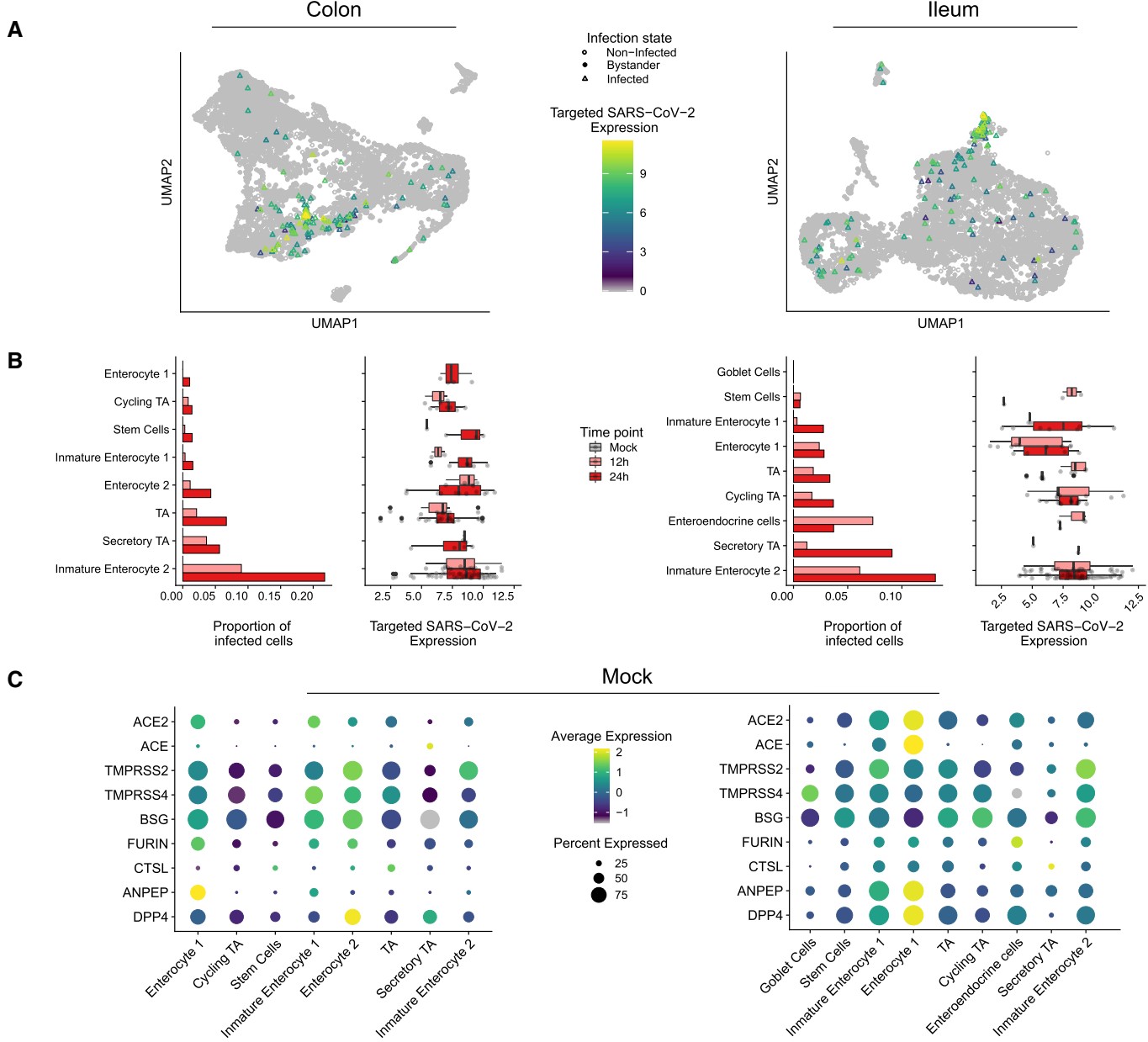

**Figure 2. SARS-CoV-2 cell tropism in human colon- and ileum-derived organoids.**

A   UMAP visualization of scRNAseq data of SARS-CoV-2-infected colon- (left) and ileum-derived organoids (right). Triangles represent infected cells and colors represent the corrected targeted normalized expression of SARS-CoV-2 determined using the targeted scRNAseq data.

B   Proportion of cells infected with SARS-CoV-2 for each cell type and corresponding boxplot of the normalized expression values of SARS-CoV-2 for each infected cell in each individual cell type in colon (left) and ileum (right) infected organoid samples. Data are color coded for mock, 12 and 24 hpi. The boxes represent the interquartile range, the horizontal line in the box is the median, and the whiskers represent 1.5 times the interquartile range. (Colon; Immature Enterocyte 2 12 h $n = 15$ & 24 h $n = 47$, Secretory TA 12 h $n = 2$ & 24 h $n = 4$, TA 12 h $n = 7$ & 24 h $n = 21$, Enterocyte 2 12 h $n = 4$ & 24 h $n = 11$, Immature Enterocyte 1 12 h $n = 2$ & 24 h $n = 7$, Stem Cells 12 h $n = 1$ & 24 h $n = 3$, Cycling TA 12 h $n = 3$ & 24 h $n = 6$, Enterocyte 1 12 h $n = 0$ & 24 h $n = 4$, Ileum; Immature Enterocyte 2 12 h $n = 31$ & 24 h $n = 50$, Secretory TA 12 h $n = 1$ & 24 h $n = 1$, Enteroendocrine cells 12 h $n = 3$ & 24 h $n = 1$, Cycling TA 12 h $n = 8$ & 24 h $n = 15$, TA 12 h $n = 5$ & 24 h $n = 5$, Enterocyte 1 12 h $n = 9$ & 24 h $n = 4$, Immature Enterocyte 1 12 h $n = 2$ & 24 h $n = 8$, Stem Cells 12 h $n = 2$ & 24 h $n = 1$).

C   Dot plots of known entry determinants across cell types. The dot size represents the percentage of cells expressing the gene; the color represents the average relative expression across the cell type. Data are from mock-infected colon (left) and ileum organoids (right).

compared to mock-infected cells (Fig 3B, right panel). Interestingly, when considering colon-derived organoids, the expression levels of these cellular genes were found slightly increased at 12 hpi and

decreased at 24 hpi (Fig 3B, left panel). Altogether, these data suggest that *ACE2* expression is downregulated in colon- and ileum-derived hIECs upon SARS-CoV-2 infection but with fundamental

differences between bystander and infected cells. These differences in the SARS-CoV-2-induced downregulation of ACE2 between colon and ileum highlight that host/pathogen interactions and mechanisms of pathogenesis can be distinct between different sections of the gastrointestinal tract. To validate this observation, we performed multiplex single-molecule fluorescence *in situ* hybridization (FISH) on SARS-CoV-2-infected organoids. At 12 and 24 hpi, organoids were fixed and evaluated using transcript-specific probes directed against the SARS-CoV-2 genome and *ACE2*. Fluorescence microscopy analysis confirmed that infected cells indeed display lower expression levels of *ACE2* at both 12 hpi and 24 hpi (Fig 3F, white arrow). Quantification of the relative expression levels of SARS-CoV-2 genome and *ACE2* transcripts in the RNA FISH images at the single-cell level again confirmed a negative correlation between SARS-COV-2 and *ACE2* (Fig 3G). Altogether, our data strongly suggest that the expression levels of *ACE2* decrease in both colon and ileum hIECs upon SARS-CoV-2 infection.

### SARS-CoV-2 induces a pro-inflammatory response in hIECs

To evaluate the response of hIECs to SARS-CoV-2 infection, we performed a comparative gene expression analysis between mock-infected and infected organoids. For the infected organoids, we considered separately the infected cells (those with SARS-CoV-2 genome detected) and the bystander cells (those without SARS-CoV-2 genome). In colon organoids, already at 12 hpi hIECs display a strong NFκB and TNF response to infection with this response becoming even more pronounced at 24 hpi (Fig EV5A and B). When comparing mock to bystander cells, we noticed that at 24 hpi, the response of bystander cells mostly followed an IFN-mediated immune response characterized by the presence of multiple ISGs (Fig EV5C and D). This observation suggests that infected cells generate a pro-inflammatory response while bystander cells likely respond to the secreted IFN in a paracrine manner. This is supported by the differential gene expression analysis of bystander vs. infected cells showing that infected cells have a stronger NFκB-

and TNF-mediated response compared with bystander cells (Fig EV5E and F). Similar results were found in SARS-CoV-2 infected ileum-derived organoids (Fig EV5G–L). Interestingly, at 24 hpi, some interferon-stimulated genes (ISGs) (*e.g.*, I*FIT1-3*, *MX1*, *CXCL10*, *IRF1*) were found to be also upregulated in infected cells but to a much lesser extent compared with bystander cells (Fig EV5G–J). Additionally, while infected cells in ileum-derived organoids were found to generate a similar NFκB/TNF-mediated response compared with the colon-derived organoids (Fig EV5B and H), ileum-derived bystander cells had a stronger IFN-mediated response which can be seen by the overall higher expression of ISGs in ileum organoids compared with colon organoids upon SARS-CoV-2 infection (Fig EV5D and J). Together, these comparative gene expression analyses revealed that upon SARS-CoV-2 infection of human intestinal epithelial cells, both strong pro-inflammatory and IFN-mediated responses are generated.

### Cell type-specific immune response in infected vs. bystander cells

Taking into account the differences in the susceptibility of the different hIEC types to SARS-CoV-2, with immature enterocytes 2 constituting the main site of SARS-CoV-2 infection (Fig 2), we compared the response of each individual cell type with SARS-CoV-2 infection. Similar to the analysis of all cell types taken together (Fig EV5), differential gene expression analysis of colon-derived infected immature enterocytes 2 revealed a strong NFκB/TNF-mediated response while bystander immature enterocytes 2 mostly display an IFN-mediated response (Fig 4A–C and Appendix Fig S3A). Similarly, in ileum-derived organoids, infected immature enterocytes 2 also showed a strong NFκB/TNF-mediated response (Fig 4F and H, and Appendix Fig S3B) while bystander cells were characterized by their response to secreted IFNs leading to ISG expression (Fig 4G and Appendix Fig S3B).

Pathway analysis confirmed that the bystander response was mostly an IFN-related response while the infected cell response was mostly pro-inflammatory (Appendix Fig S4). Comparison of the

▶

**Figure 3. Downregulation of ACE2 upon SARS-CoV-2 infection.**

A Volcano plots of genes that are differentially expressed in infected cells relative to mock-infected cells at 12 hpi and 24 hpi in ileum organoids. The statistical significance (-log10 of the adjusted *P*-value) is shown as a function of the log2 fold change. MAST tests were used to generate *P*-values, bonferroni multiple hypotheses correction was used to compute *FDR values*. Labeled dots in all panels are gene names of selected differentially expressed genes between the compared two populations.

B Dot plots displaying the expression changes of known SARS-CoV-2 entry determinants for both infected and bystander cells during the course of infection (mock, 12 and 24 hpi) in colon (left) and ileum organoids (right). The dot size represents the percentage of cells expressing the gene; the color represents the average relative expression across the cell type excluding zeros.

C Pearson correlation of gene expression values with the amount of SARS-CoV-2 genome (y-axis) vs the maximal log2 fold change (x-axis) across conditions. This plot is generated by comparing both 12 and 24 hpi to mock. Top 5 high correlated and anticorrelated genes and known SARS-CoV-2 entry determinants are highlighted in blue.

D Correlation of SARS-CoV-2 expression with *ACE2* expression across all infected cell types from ileum organoids at 24 hpi. (*n* = 146) Gray area represents the standard error to the fitted line.

E Global-scaled log normalized expression values of *ACE2* in each cell type for mock-infected and SARS-CoV-2-infected cells in ileum organoids at 12 hpi and 24 hpi. The boxes represent the interquartile range, the horizontal line in the box is the median, and the whiskers represent 1.5 times the interquartile range (Immature Enterocyte 2 Mock *n* = 377, 12 h *n* = 390 & 24 h *n* = 270, Stem Cells Mock *n* = 184, 12 h *n* = 219 & 24 h *n* = 115, TA Mock *n* = 248, 12 h *n* = 241 & 24 h *n* = 128, Immature Enterocyte 1 Mock *n* = 452, 12 h *n* = 559 & 24 h *n* = 280, Cycling TA Mock *n* = 187, 12 h *n* = 240 & 24 h *n* = 214, Enterocyte 1 Mock *n* = 249, 12 h *n* = 374 & 24 h *n* = 140, Goblet Cells Mock *n* = 12, 12 h *n* = 24 & 24 h *n* = 5, Secretory TA Mock *n* = 7, 12 h *n* = 59 & 24 h *n* = 2, Enteroendocrine cells Mock *n* = 26, 12 h *n* = 30 & 24 h *n* = 15).

F Multiplex *in situ* RNA hybridization of *ACE2* and SARS-CoV-2 of mock-infected and infected 2D ileum organoids at 12 hpi and 24 hpi. White arrows point at SARS-CoV-2-infected cells. A representative image is shown.

G Correlation of the relative expression SARS-CoV-2 and *ACE2* for infected cells from the multiplex *in situ* RNA hybridization shown in (F). Each dot represents a cell.

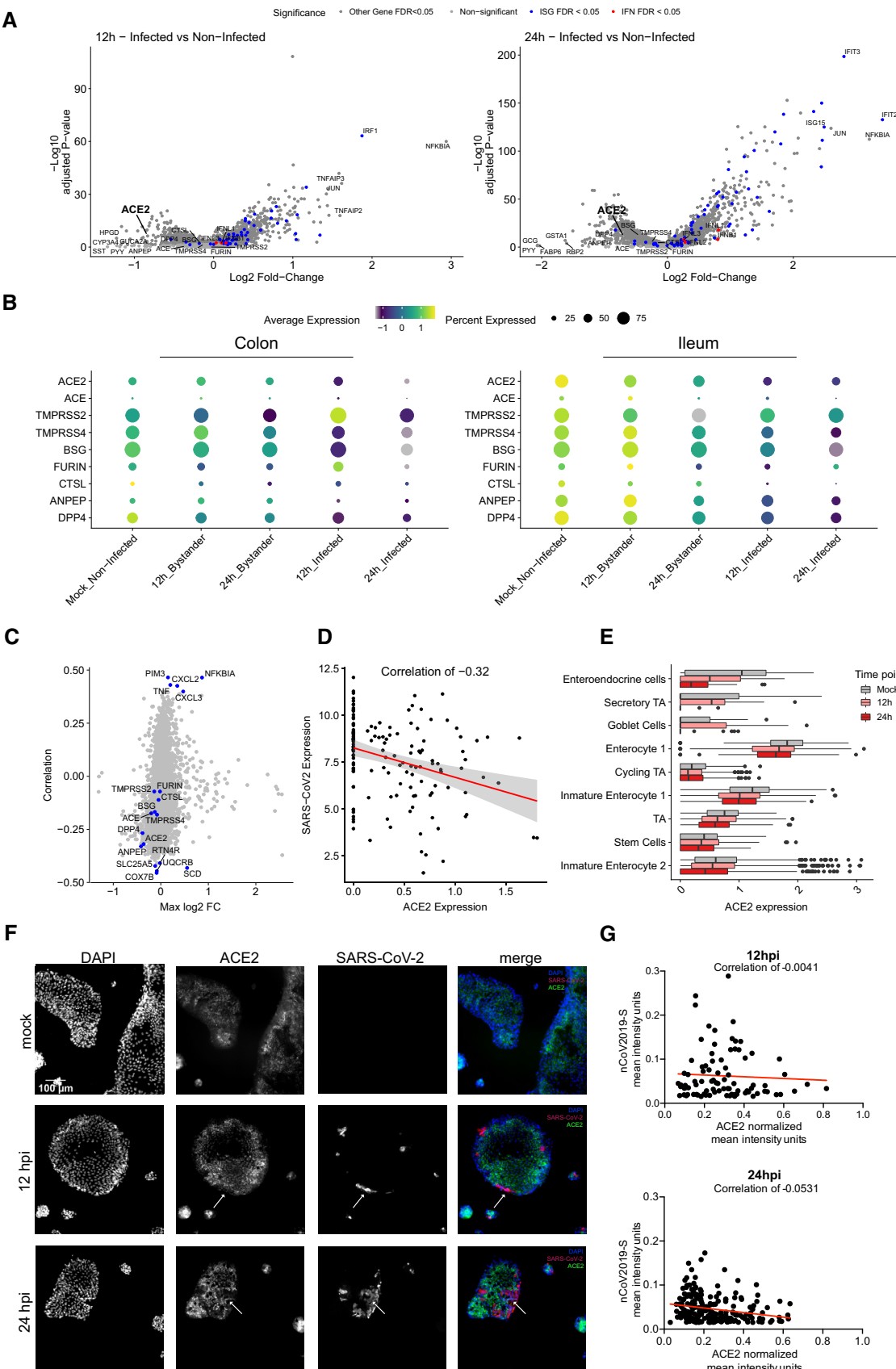

Figure 3.

transcriptional response to SARS-CoV-2 infection in infected *vs.* bystander immature enterocytes 2 further confirmed that infected cells mount a strong pro-inflammatory response characterized by the upregulation of NFκB and TNF (Fig 4C and D, and H and I). Analysis of the top 30 differentially expressed ISGs in colon-derived organoids clearly shows that at 24 hpi, bystander cells respond to IFN by upregulating the expression of a large panel of ISGs (Fig 4E). Similar findings were observed in immature enterocytes 2 from ileum organoids, although infected cells were also found to express more ISGs compared with their colon-derived counterparts.

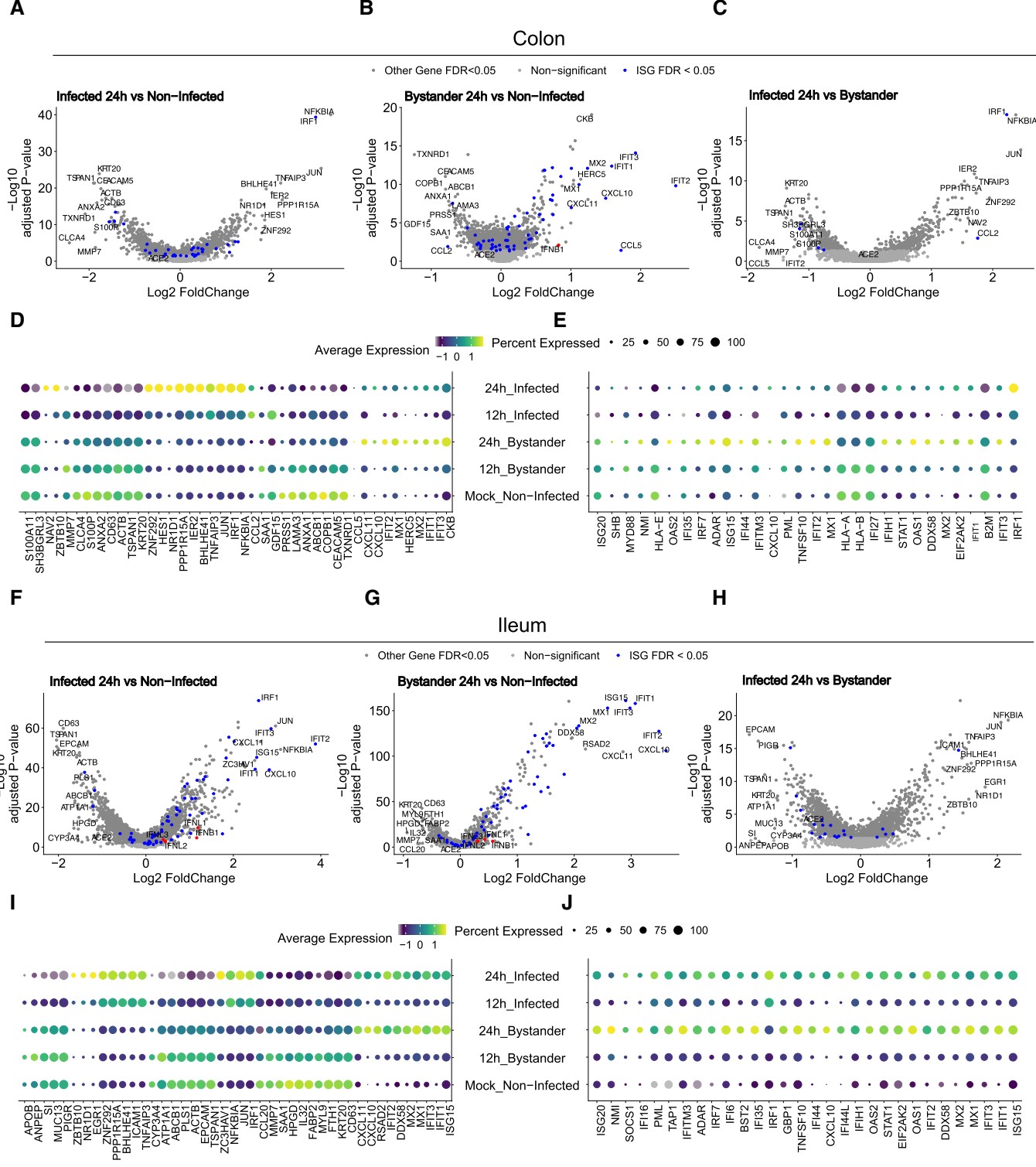

**Figure 4.**

**Figure 4. Intrinsic innate immune response generated by immature enterocytes 2 upon SARS-CoV-2 infection.**

A–C  Volcano plots displaying the genes that are differentially expressed in immature enterocytes 2 upon SARS-CoV-2 infection of colon organoids. (A) infected *vs*. mock-infected cells, (B) bystander *vs*. mock-infected cells and (C) infected *vs*. bystander cells. The statistical significance (-log10 adjusted *P*-value) is shown as a function of the log2 fold change. MAST tests were used to generate *P*-values, bonferroni multiple hypotheses correction was used to compute *FDR values*. Labeled dots in blue in all panels are gene names of selected differentially expressed genes between the compared two populations. Labeled dots in red in all panels are gene corresponding to interferon if detected.

D  Dot plot of the top 42 most differentially expressed genes upon SARS-CoV-2 infection in mock, infected and bystander cells at 12 and 24 hpi. The dot size represents the percentage of cells expressing the gene; the color represents the average relative expression across the cell type.

E  Same as in (D) but for the top 30 most differentially expressed ISGs.

F–J  Same as (A–E) but for ileum organoids. Labeled dots in blue in all panels are gene names of selected differentially expressed genes between the compared two populations. Labeled dots in red in all panels are gene corresponding to interferon if detected.

Importantly, in both colon and ileum organoids, bystanders showed higher levels of expression for all considered ISGs compared with infected cells (Fig 4E and J). These results are consistent with the observation that ileum-derived organoids are more immune-responsive compared with colon-derived organoids (Fig EV5D and J). Together these data show that upon SARS-CoV-2 infection, infected cells mount a strong NFκB/TNF-mediated pro-inflammatory response and display a limited production of ISGs while bystander cells mount a strong IFN-mediated response through a strongly upregulated expression of a broad panel of ISGs.

To determine whether the characterized NFκB/TNF-high and IFN-low immune response is specific to immature enterocytes 2, we also looked individually at each infected cell type. While it is hard to draw strong conclusion on how infected cells behave for all cell types due to their low permissivity to SARS-CoV-2 infection (low number infected cells) and low level of viral transcripts, we can observe that infected cells display a strong NFκB/TNF-mediated response (Appendix Figs S5–S8). Interestingly, we observed that bystander cells from different cell types produce different patterns of ISGs (Appendix Figs S5–S8). Since our targeted scRNAseq analysis revealed the presence of background viral RNA in the samples (Fig EV3), we asked whether the observed immune response is indeed cascading to bystander cells through type III interferon secreted by infected cells or is caused by the direct action of non-replicating viral particles on bystander cells. To address this, colon and ileum organoids were infected with either live or UV-inactivated SARS-CoV-2. Results revealed that upon infection with live SARS-CoV-2 both IFNs and ISGs were produced (Appendix Fig S9). On the contrary, exposure of organoids to UV-inactivated SARS-CoV-2 did not lead to virus replication and cells failed to produce both IFN and ISGs (Appendix Fig S9). This demonstrates that active replication is required for the described immune response and allowed us to rule out the exposure to non-replicating viral particles as being the cause of this response. Altogether, our results show that infected and bystander cells respond differently to SARS-CoV-2 infection where infected cells mount a NFκB/TNF-mediated response while bystander cells mount a IFN-mediated response.

### Signaling activity in infected vs. bystander cells

To characterize the signaling that underpins the distinct immune response of infected and bystander cells upon SARS-CoV-2 infection, we inferred the pathway signaling activity from scRNAseq data with PROGENy (Fig 5A and B). For both colon and ileum organoids, infected cells show a strong activation of the MAPKs, NFκB, and TNFα pathways. In line with the enrichment analysis (Appendix Fig

S3), these pathways were found to be less activated in bystander cells with higher scores in ileum compared with colon (Fig 5A and B). Interestingly and in accordance with our differential gene expression analyses (Figs 4 and EV5 and Appendix Fig S3), the JAK-STAT signaling pathway was found to be activated mostly in bystander cells (Fig 5A and B). To further elucidate the signaling activity at the single-cell level, we generated diffusion maps of all single cells based on the scRNAseq expression of interferon-related genes (Fig 5C and D). In both ileum and colon, we observed a clear bifurcation of all cells into two distinct branches, one branch representing mainly infected cells and another branch representing mainly bystander cells. Calculation of transcription factor activities (TFA) based on the gene expression of their target can provide a more robust measure of the effect of transcription factors (TFs) when compared to only gene expression. Hence, we calculated the TFA of selected transcription factors for all single cells using SCENIC and mapped the inferred activities onto the single-cell diffusion maps (Fig 5C and D, right insets). We found that the transcription factors STAT1 and IRF1 were activated mainly in bystander cells (branch along DC1) while JUN was activated in infected cells (branch along DC2) (Fig 5C and D, left panels). Extending this analysis to transcription factors whose activity pattern is highly correlated to either DC1 or DC2 revealed that globally, transcription factors that are critical for IFN-mediated signaling (*i.e.* the ISGF3 complex: STAT1/STAT2/IRF9 and IRF1) are highly active in bystander cells (Fig 5E and F). Similarly, the ETS variant transcription factor 7 (ETV7) which is an ISG acting as a negative regulator of IFN-mediated signaling (preprint: Froggatt *et al*, 2019) and the Early Growth Response Gene 1 (EGR1) which enhances type I IFN signaling (Zhu *et al*, 2018) were also found to be activated in bystander cells (Fig 5E and F). Upregulation of the EGR1- and JUN-dependent pathways was consistent with the findings of the previous work investigating SARS-CoV-2 infection of the human lung and intestinal epithelial cell lines Calu-3 and Caco-2 (preprint: Wyler *et al*, 2020).

### SARS-CoV-2 inhibits IFN-mediated ISG expression

To validate that the IFN-mediated response is specific to bystander cells, ileum-derived organoids were infected with SARS-CoV-2. At 12 and 24 hpi, single-molecule RNA FISH was performed using probes specific for the SARS-CoV-2 genome and for *ISG15* which was found to be highly upregulated upon infection and has the highest -log10 *P*-value in the differential analysis comparing bystander cells vs mock-infected cells (Fig 4G). Microscopy images revealed that bystander cells (non-infected) were indeed positive for *ISG15*

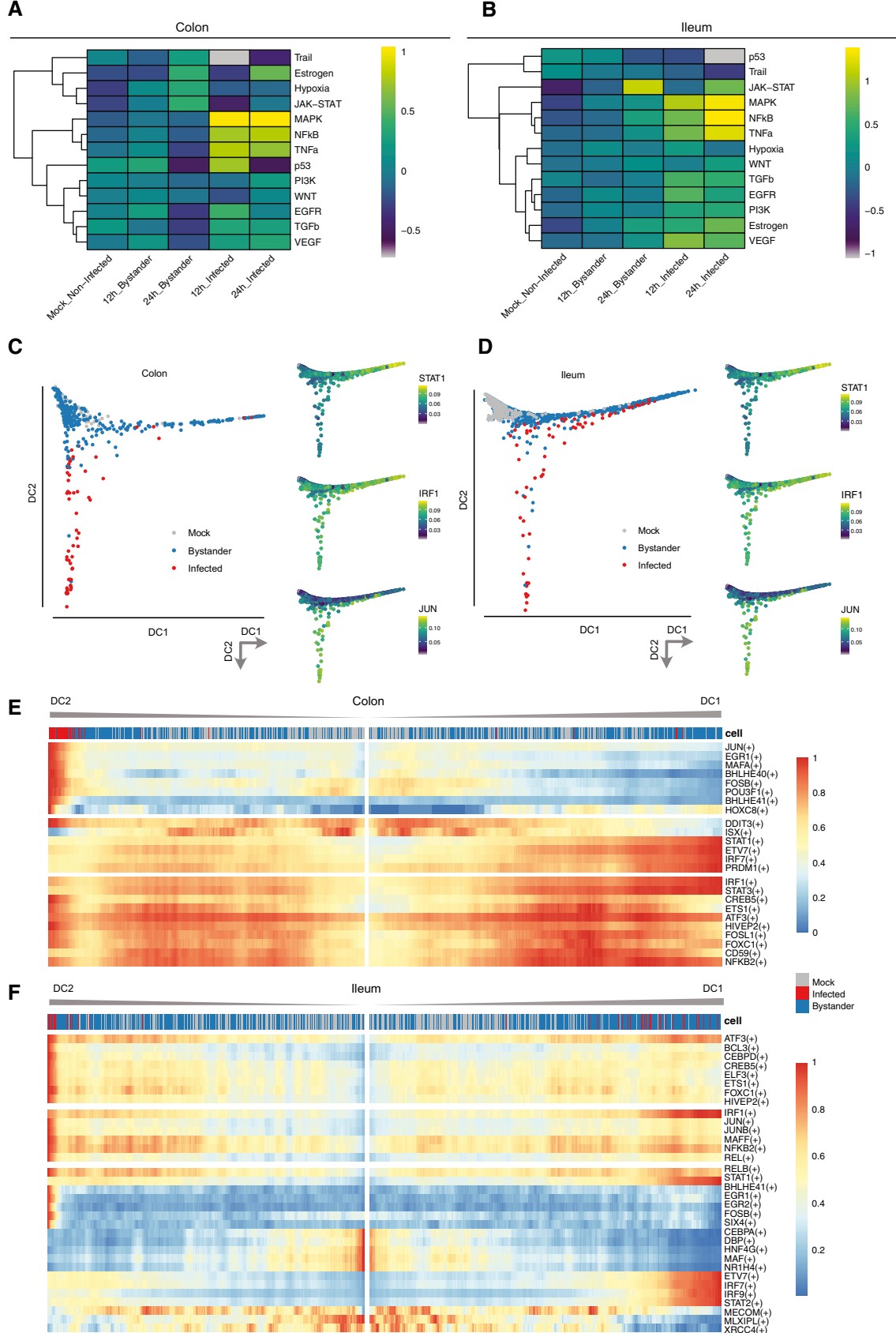

**Figure 5.**

**Figure 5. Differential signaling activity of infected *vs*. bystander cells upon SARS-CoV-2 infection.**

A, B   Heatmaps of scaled relative pathway signaling activity inferred by PROGENy for (A) colon and (B) ileum organoids.
C, D   Diffusion maps embeddings showing mock, bystander, and infected cells (big panels) and activities for selected transcription factors STAT1, IRF1, and JUN (small panels) for (C) colon and (D) ileum organoids. Case color scales represent absolute score values in arbitrary units.
E, F   Heatmaps of transcription factor activities along bifurcating trajectories of the corresponding diffusion maps for (E) colon and (F) ileum organoids. Column top annotations represent the infection status of each cell (infected, bystander, and mock). The dimensions DC1 and DC2 represent the first two eigenvectors of the Markovian transition matrix and were calculated separately for either colon or ileum organoids. Case color scales represent absolute score values in arbitrary units.

(Fig 6A). Interestingly, SARS-CoV-2 infected cells were found to express little to no *ISG15* (Fig 6A, white arrows). Quantification confirmed that cells which displayed the highest expression of ISG15 were negative for SARS-CoV-2 genome (Fig 6B). These results support the model that bystander cells respond to SARS-CoV-2 infection by mounting an IFN-mediated response. On the other hand, as shown by using both scRNAseq (Fig 4) and RNA FISH (Fig 6A), infected cells do not mount an IFN-mediated response (low-to-no expression of ISGs) suggesting that infected cells are refractory to IFN. To address whether SARS-CoV-2 infection can indeed impair IFN-mediated signaling, we directly monitored ISG production in response to IFN treatment in either infected or non-infected cells. However, while it is well known that ISGs are made in a JAK-STAT-dependent manner downstream the IFN receptors, there is growing evidence that a subset of ISGs can be made downstream IRF3 following viral recognition by PRRs. To alleviate this unwanted IRF3 mediated production of ISGs, we generated a human intestinal cell line (T84) depleted of *IRF3*. Infection of the *IRF3* KO T84 did not result in the production of ISGs as monitored by q-RT–PCR of *ISG15* (Fig 6D). *IRF3* KO T84 cells were mock-infected or infected with SARS-CoV-2, at 24 hpi cells were treated with IFN and 12 h post-treatment production of ISG15 was assessed by q-RT–PCR and normalized to the housekeeping gene (TBP) (Fig 6C). Results show that mock-infected IRF3 KO T84 cells were responsive to IFN demonstrating that genetic depletion of *IRF3* does not alter IFN-mediated signaling (Fig 6D). Interestingly, in SARS-CoV-2 infected IRF3 KO T84 cells, production of *ISG15* upon IFN treatment was significantly downregulated (Fig 6D). To confirm that this impaired induction of ISG15 upon IFN treatment was specific to SARS-CoV-2 infection, IRF3 KO T84 cells were infected with astrovirus at an MOI of 3 to achieve full infection (Fig 6E). Contrary to SARS-CoV-2 infection, infection of *IRF3* KO T84 cells by astrovirus did not impair IFN-mediated signaling as a similar upregulation of *ISG15* was observed in both mock-infected and astrovirus-infected cells upon IFN treatment (Fig 6D). In a second validation approach, to fully demonstrate that only infected cells have an altered IFN-mediated signaling, we developed an assay based on a fluorescent reporter of ISG expression (Fig 6F). For this, we generated a T84 cell line transduced with a reporter made of the promoter region of the ISG *MX1* driving the expression of the fluorescent protein mCherry. *Mx1* is known to be made strictly downstream of the IFN receptor in a JAK-STAT-dependent manner but not downstream of IRF3. Upon IFN treatment, about 30–40% of cells expressing this reporter were responsive and became fluorescent (Fig 6G). Following infection with SARS-CoV-2 at a multiplicity of infection (MOI) of 3, most of the cells were found to be infected (Fig 6G). However, when cells were treated 24 hpi with IFN, most infected cells did not respond to IFN and very few became double-positive for both virus and *MX1* driven mCherry (Fig 6G, left panel). To control that non-infected

bystander cells could indeed respond to IFN and express mCherry, we repeated this experiment but using an MOI of 1 for SARS-CoV-2 infection. About 40% of the cells were found to be infected. Supplementing IFN affected mainly non-infected cells, as can be seen from the increase in MX1-positive cells and no change in the number of double-positive cells (both infected and MX1-positive) (Fig 6G, right panel).

Altogether, our results provide a strong evidence that upon infection with SARS-CoV-2 primary human intestinal cells generate a strong NFκB/TNF-mediated response and produce IFN. This IFN acts in a paracrine manner onto bystander cells that leads to the upregulation of IFN-stimulated genes. Importantly, SARS-CoV-2 infection renders infected cells refractory to IFN as they show little-to-no increase in the activity of the JAK-STAT signaling pathways and fail to upregulate IFN-stimulated gene expression.

## Discussion

Many COVID-19 patients display gastroenteritis symptoms and there is growing evidence that the intestinal epithelium can be infected by SARS-CoV-2. Whether the symptoms are associated with the direct replication of SARS-CoV-2 in the GI tract or are a consequence of the strong pro-inflammatory response seen in patients is unclear. The use of human intestinal "mini-gut" organoids has already demonstrated that intestinal epithelium cells can support SARS-CoV-2 infection, replication, and spread. However, which cell type is infected remains poorly defined (Lamers *et al*, 2020; Stanifer *et al*, 2020b; Zang *et al*, 2020). By exploiting single-cell transcriptomics approaches (scRNAseq) and targeted scRNAseq, we identified that a subpopulation of enterocytes (namely, immature enterocytes 2) is the cell type most susceptible to SARS-CoV-2 infection. We also characterized the cell type-specific response to SARS-CoV-2 infection and distinguished how bystander cells respond compared with infected cells. A visual schematic of our key findings can be found in Fig 7. Interestingly, other cell types also supported infection by SARS-CoV-2 but to a much lesser extent (Fig 2B). The characterized tropism of SARS-CoV-2 could be explained by either cell type-specific intrinsic differences rendering some cell type more permissive or due to an overrepresentation of cells of a particular cell type. In our colon-derived organoids, there were twice as many immature enterocytes 1 compared with immature enterocytes 2 and in ileum-derived organoids both enterocyte lineages were present in roughly equal numbers. This suggests that the SARS-CoV-2 cell tropism for immature enterocytes 2 is not due to a higher proportion of these cells in our organoids but due to intrinsic differences between immature enterocytes 2 and other epithelial cell lineages. Differential gene expression analysis between immature enterocytes 2 and the most similar other annotated cell type (immature enterocytes 1) does not

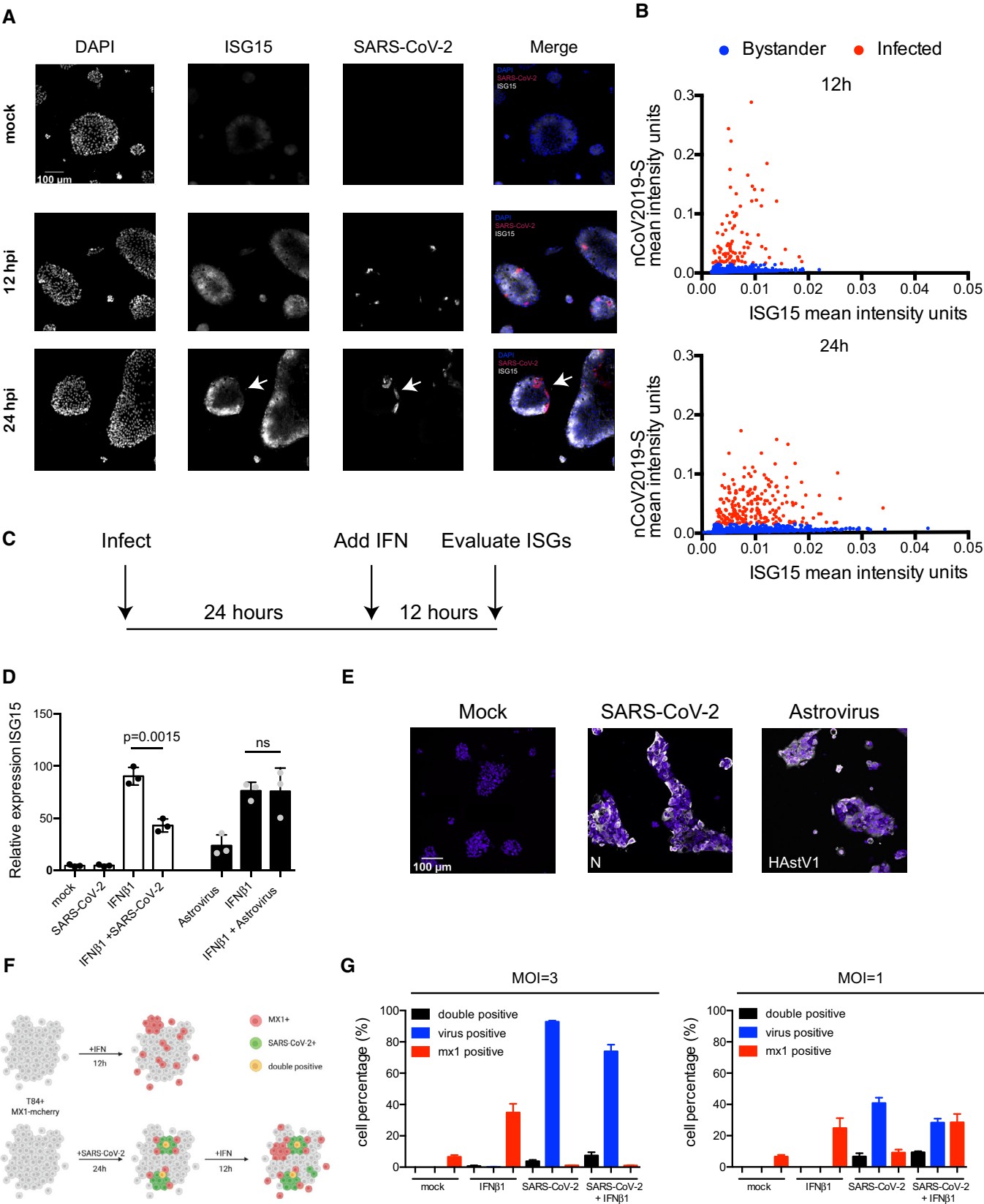

**Figure 6.**

◄

**Figure 6.  SARS-CoV-2 infection impairs interferon-mediated signaling.**

A    Human ileum-derived organoids were seeded in 2D on iBIDi chamber slides. 12 and 24 hpi cells were fixed and the amount of SARS-CoV-2-infected cells (red) and the induction of *ISG15* (white) was analyzed by single-molecule RNA FISH. Nuclei were visualized with DAPI (blue). White arrows indicate SARS-CoV-2-positive cells. *N* = 3 biological samples. Representative image is shown. Scale bar=100 um.
B    Quantification of the mean fluorescence intensity of samples in (A). Each dot represents a cell.
C    Schematic depicting the infection and interferon addition to evaluate ISG15 induction following SARS-CoV-2 and astrovirus infection.
D    T84 *IRF3* knockout cells were infected with SARS-CoV-2 or human astrovirus 1 at an MOI=3. 24 hpi, cells were incubated in the presence or absence of 2,000 IU/ml of IFNβ1. 12 h post-treatment RNA was harvested and the induction of *ISG15* was analyzed by q-RT–PCR and normalized to the housekeeping gene TBP. *N* = 3 biological replicates. Error bar indicates standard deviation. Statistics were determined by unpaired t-test.
E    T84 IRF3 knockout cells were infected with SARS-CoV-2 or human astrovirus 1 at an MOI=3. 24 hpi, cells were fixed and stained for either the SARS-CoV-2 N protein or for human astrovirus 1 (HAstV1) capsid protein. *N* = 3 biological replicates. Representative image is shown. Scale bar=100 um.
F    Schematic depicting the T84 MX1 mcherry expressing cells and their response to IFN or SARS-CoV-2 infection.
G    T84 MX1 mcherry were infected with SARS-CoV-2 at an MOI=3 or MOI=1. 24 hpi cells were incubated in the presence or absence of 2,000 IU/ml of IFNβ1. 12 h post-treatment cells were fixed and stained for SARS-CoV-2 N protein. The number of MX1 positive, SARS-CoV-2 positive, and double-positive cells was quantified. *N* = 3 biological replicates were performed. Error bar indicates standard deviation.

highlight the presence or absence of obvious restriction/replication factors that could explain the observed tropism. Interestingly, when looking at the ISG expression patterns of the different cell types present in colon and ileum organoids (Appendix Figs S5–S8, respectively), we could observe that different cell types make different ISGs and this might participate in permissiveness of cell types and as such viral tropism.

Intestinal organoids have become very important human intestinal epithelium models to study development and establishment of cell lineages (Sato *et al*, 2011). Their use in infectious disease research has also become apparent over the past years particularly with the SARS-CoV-2 pandemic (Lamers *et al*, 2020; Stanifer *et al*, 2020b; Zang *et al*, 2020). While organoids allow us to look at cell type-specific response to various challenges, they have their limitations. In the case of intestinal organoids, while most lineages are present, the ratio of the various cell types is not identical to the

proportion observed in vivo (Fujii *et al*, 2018; preprint: Triana *et al*, 2020). Additionally, comparison of the transcription profiles of each intestinal cell type revealed that the signature of gene expression of cells in an organoid context is slightly different compared with the same cell type in its physiological tissue environment (Fujii *et al*, 2018; preprint: Triana *et al*, 2020). The origin for these differences is not yet fully understood but they are likely a consequence of the methodology used to grow and maintain organoids. First, intestinal organoids are grown and maintained in a high Wnt media which keeps them constantly proliferating which is very different from the normal environment found within the body (Sato *et al*, 2011). Second, to induce differentiation, media conditions are changed and these conditions must be slightly adapted to promote differentiation toward specific cell types (Boonekamp *et al*, 2020; Ding *et al*, 2020). Most importantly, organoids are lacking the environmental features of the body (e.g., microbiota, immune cells, hypoxia) which is likely

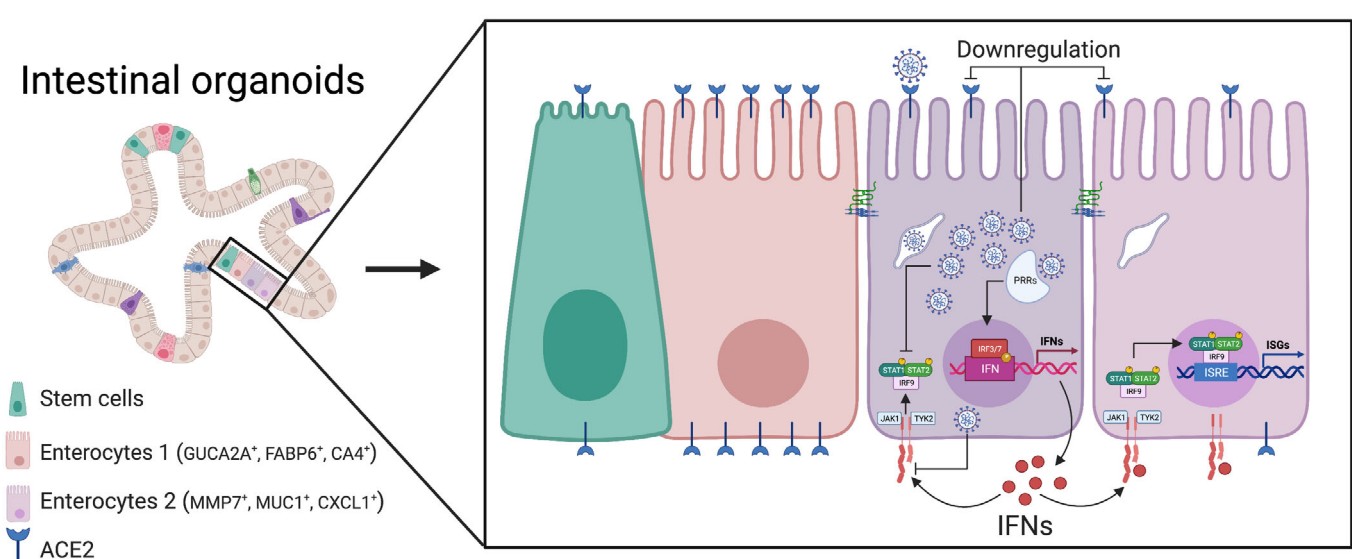

**Figure 7.  Schematic of SARS-CoV-2 infection of human intestinal epithelial cells.**

SARS-CoV-2 infects a subpopulation of enterocytes in human mini-gut organoids. Upon infection, enterocytes mount a pro-inflammatory response characterized by the upregulation of NFκB and TNF. Bystander cells respond to secreted IFN and upregulate the expression of ISGs. SARS-CoV-2 infection induces the downregulation of *ACE2* expression and interferes with IFN-mediated signaling in infected cells. Created with BioRender.

responsible for fundamental differences in both how stem cell differentiates in all intestinal lineages and in the cell type-specific function/activity of intestinal epithelial cells.

The expression levels of the SARS-CoV-2 receptor *ACE2* were found to be higher in immature enterocytes 1 while immature enterocytes 2 express more of the key cellular protease *TMPRSS2* (Figs 2C and EV5). Although expression of ACE2 is mandatory for infection (Hoffmann *et al*, 2020), we noticed no correlation between *ACE2* expression level and the copy numbers of SARS-CoV-2 genome in the cell (Fig 2B and C). This observation is important as it raises a question about using *ACE2* expression as the only basis for conjectures about infectability of cell types or even organs, the approach recently proposed in various SARS-CoV-2-related publications (preprint: Han *et al*, 2020; Lukassen *et al*, 2020; Qi *et al*, 2020; preprint: Ravindra *et al*, 2020; Singh *et al*, 2020; Sungnak *et al*, 2020; Zhao *et al*, 2020; preprint: Zhou *et al*, 2020b). Our results highlight that the investigation of SARS-CoV-2 tropism requires biological validation of infection and should not be done solely based on analysis of transcriptional profiles of individual cells or tissues. Interestingly, we found that *TMPRSS2* expression levels were well associated with the SARS-CoV-2 genome copy numbers in human intestinal epithelial cells (Figs 2C and EV4) which is consistent with the observation that TMPRSS2 and the related protease TMPRSS4 are critical for infection of intestinal organoids (Zang *et al*, 2020). As such, it is tempting to speculate that TMPRSS2 plays a more important role in the SARS-CoV-2 cell tropism than ACE2; however, more studies are required to fully explore this hypothesis.

Several studies have suggested that *ACE2* is an interferon-stimulated gene and is induced upon SARS-CoV-2 infection (Chua *et al*, 2020; Ziegler *et al*, 2020). This led to a speculation that upon infection and induction of pro-inflammatory responses, the *ACE2* levels would increase thereby favoring SARS-CoV-2 infection. Our results show that, on the contrary, upon infection *ACE2* levels decrease both in infected and bystander hIECs (Figs 3 and 7). Interestingly, differences in the kinetics of *ACE2* regulation were observed between colon- and ileum-derived organoids. In colon-derived organoids, a small increase in *ACE2* expression was observed at early times post-infection (12 hpi) but at later time points (24 hpi) the overall expression of *ACE2* (and other putative SARS-CoV-2 receptors and key cellular proteases) was decreased as compared to mock-infected cells (Fig 3B). In ileum organoids, the expression of *ACE2* was decreased over the time course of infection. The observed upregulation of *ACE2* upon infection might be tissue-specific and time-dependent. However, recently it has been proposed that ACE2 does not behave as an ISG but instead a novel form of *ACE2* (*dACE2*) is interferon-inducible (Onabajo *et al*, 2020). *dACE2* results from transcription initiation at an internal exon leading to the production of an alternative short version. Within our scRNAseq data, we could not distinguish between the two forms and as such the observed temporal increase (in colon-derived organoids) could be due to the downregulation of *ACE2* with the concomitant upregulation of *dACE2*.

The nature of the PRR responsible for sensing SARS-CoV-2 infection is yet to be determined but from recent work and previous work on SARS-CoV-1 and MERS it could be speculated that TLR3, RLRs, and the STING pathways could be involved (preprint: Neufeldt *et al*, 2020; Park & Iwasaki, 2020). In our current work, we could show that active virus replication is required to induce an IFN-mediated response as infection by UV-inactivated SARS-CoV-2 did not lead to IFN and ISG production (Appendix Fig S9). Interestingly, when human intestinal epithelial cells are infected by a UV-inactivated reovirus, which is a virus whose genome is a dsRNA, an IFN-mediated response is induced (Stanifer *et al*, 2016). As SARS-CoV-2 is single-stranded RNA virus and dsRNA intermediates will only occur during active replication, it is tempting to speculate that what is being sensed are these dsRNA replication intermediates.

SARS-CoV-2 infection is characterized by a strong pro-inflammatory response, and this has been observed both in tissue-derived samples and *in vitro* using cell culture models (Blanco-Melo *et al*, 2020). This pro-inflammatory response is characterized by upregulation of the NFκB and TNF pathways. Our scRNAseq approach revealed that this pro-inflammatory response is specific to infected cells and that bystander cells do not show a strong pro-inflammatory response. These differences between infected and bystander cells were earlier observed for other cell types: Infection of human bronchial epithelial cells (HBECs) also reveal that the pro-inflammatory response is biased toward infected cells and not bystander cells (preprint: Ravindra *et al*, 2020).

It was reported that infection of human lung epithelial cells by SARS-CoV-2 is characterized by a low to absent IFN response (Blanco-Melo *et al*, 2020). On the contrary, in human intestinal epithelium cells an IFN-mediated response is readily detectable and is characterized by both the production of IFN and ISGs (Lamers *et al*, 2020; Stanifer *et al*, 2020b). Interestingly, upon infection with SARS-CoV-2, we observed the upregulation of *IFNλ2-3* but we failed to observe a significant increase in *IFNλ1* expression (Fig EV1H). This absence of IFNλ1 upregulation is not specific to SARS-CoV-2 but a particularity of intestinal organoids, as a similar *IFNλ2-3*-specific response was observed when intestinal organoids were infected with other enteric viruses (Pervolaraki *et al*, 2017; Stanifer *et al*, 2020c). Upregulation of IFN production upon SARS-CoV-2 infection of intestinal epithelial cells was found to be low but significant (Fig 4), and this could raise the question whether this small production of IFN is sufficient to induce the production of ISGs in a paracrine manner. Previous work on the antiviral function in type III IFN in human intestinal epithelial cells revealed that although type III IFN protects epithelial cells against infection in a dose-dependent manner, very small amounts of IFN are required to induce ISGs and to provide an antiviral state to the cells (Pervolaraki *et al*, 2017). The current work confirms this model as despite a moderated upregulation of IFN expression in infected cells, we observed a strong ISG upregulation in bystander cells.

When comparing the immune response generated by organoids derived from different sections of the GI tract, we observed that ileum organoids were more immunoresponsive compared with colon organoids. While the extent of the upregulation of genes related to pro-inflammatory response was similar between colon and ileum (Fig 4A and F), we observed that ileum organoids, particularly bystander cells produced significantly more ISGs compared with their colon counterparts (Fig 4B and G). This compartmentalization of response along the GI tract is consistent with previous reports describing that different sections of the GI tract respond differently to microbial challenges (Kayisoglu *et al*, 2020). Most importantly, our results reveal that production of ISGs is mostly restricted to bystander cells, while production of IFN is detected mostly in infected cells (Figs 5 and EV5, and Appendix Fig S3). These observed differences between

infected and bystander cells were confirmed using single-molecule RNA FISH showing that production of *ISG15* was clearly observed in bystander and, to a much lesser extent, in infected cells (Fig 6). An important finding of this work is that infected cells not only fail to produce ISGs, but they also become refractory to IFN (Figs 6 and 7). When SARS-CoV-2-infected intestinal cells were treated with IFN, only bystander cells upregulated ISG while infected cells did not. This absence of ISG induction in infected cells suggests that SARS-CoV-2 has developed mechanisms to shutdown IFN-mediated signaling and the subsequent production of ISGs (Fig 7). Preventing IFN-mediated signaling in infected cells would provide a replication advantage to SARS-CoV-2 as secreted IFN will not be able to act in an autocrine manner to induce ISGs which will curtail virus replication and *de novo* virus production. Although the SARS-CoV-2 viral protein responsible for blocking the IFN-mediated signaling is yet to be identified in our system, a recent report suggests that ORF6 could block IFN-mediated signaling by interfering with STAT1 nuclear translocation (Lei *et al*, 2020).

In conclusion, in this work we identified a subset of immature enterocytes as the primary site of infection of SARS-CoV-2 in ileum- and colon-derived human intestinal epithelial cells. We could show that upon infection, infected cells mount a strong pro-inflammatory response characterized by a strong activation of the NFκB/TNF pathways while bystander cells mount an IFN-mediated response (Fig 7). This differential response between infected and bystander cells is due to an active block of IFN signaling in infected cells (Fig 7). Although our work was performed in primary non-transformed human intestinal epithelial, it will be important to validate our findings (SARS-CoV-2 tropism, down regulation of ACE2, and inhibition of IFN response) in a tissue from infected patients as the physiological gut environment may modify host/pathogen interactions globally and/or in a cell type-specific manner. Interestingly, recent work performing scRNAseq of SARS-CoV-2 infected HBECs revealed that infected cells were readily responding to secreted interferon and produced large amounts of ISGs (preprint: Ravindra *et al*, 2020). This suggests that there are cell type-specific or tissue-specific regulations of interferon-mediated signaling during SARS-CoV-2 infection. This needs to be considered when studying replication and pathogenesis of SARS-CoV-2 in different organs as well as when developing therapies against COVID-19.

# Materials and Methods

## Reagents and Tools Table

| Reagent/Resource | Reference or Source | Identifier or Catalog Number |
|---|---|---|
| **Experimental Models** | | |
| Human colon organoids | This paper | n/a |
| Human ileum organoids | This paper | n/a |
| Vero E6 cells | ATCC | CRL 1586 |
| L-WRN cells | ATCC | CRL-3276 |
| T84 WT cells | ATCC | CCL-248 |
| T84 IRF3 KO cells | This study | n/a |
| T84 MXI-cherry cells | This study | n/a |
| Caco-2 cells | ATCC | HTB-37 |
| SARS-CoV-2 BavPat1/2020 | European Virology Archives | 026V-03883 |
| Human astrovirus 1 | Gift from Stacy Schultz-Cherry | n/a |
| **Recombinant DNA** | | |
| MX1-cherry | Gift from Ronald Dijkman | University of Bern |
| lentiCRISPR v2 | Addgene | #52961 |
| **Antibodies** | | |
| Mouse monoclonal anti-SARS CoV NP | Sino Biologicals | Cat#MM05 |
| Mouse monoclonal anti-dsRNA (J2) | Scions | Cat#10010200 |
| Mouse monoclonal anti-Human Astrovirus 1 (8E7) | Thermo Fischer Scientific | Cat#MA5-16293 |
| Alexa Fluor Goat anti mouse 488 | Thermo Fischer Scientific | Cat#A-11001 |
| Alexa Fluor Goat anti rabbit 488 | Thermo Fischer Scientific | Cat#A-11034 |
| Alexa Fluor Goat anti mouse 568 | Thermo Fischer Scientific | Cat#A-21124 |
| Alexa Fluor Goat anti rabbit 568 | Thermo Fischer Scientific | Cat#A-11011 |
| Anti-Mouse IgG IRDye CW800 | LiCor | Cat#926-32212 |

                                

**Reagents and Tools table**  (continued)

| Reagent/Resource | Reference or Source | Identifier or Catalog Number |
|---|---|---|
| **Oligonucleotides and other sequence-based reagents** | | |
| IRF3 KO guide RNA1 | Eurofins | 5' CACCGGAGGTGACAGCCTTCTACCG 3'<br>5' AAACCGGTAGAAGGCTGTCACCTCC 3' |
| IRF3 KO guide RNA2 | Eurofins | 5' AAACGCTGCGACACCAGCCAGGGCC 3'<br>5' CACCGGCCCTGGCTGGTGTCGCAGC 3' |
| IRF3 KO guide RNA3 | Eurofins | 5' CACCGTGCACCAGTGGCCTCGGCCC 3'<br>5' AAACGGGCCGAGGCCACTGGTGCAC 3' |
| TBP forward | Eurofins | CCACTCACAGACTCTCACAAC |
| TBP reverse | Eurofins | CTGCGGTACAATCCCAGAACT |
| IFNB1 forward | Eurofins | GCCGCATTGACCATCTAT |
| IFNB1 reverse | Eurofins | GTCTCATTCCAGCCAGTG |
| IFNL1 forward | Eurofins | GCAGGTTCAAATCTCTGTCAC |
| IFNL1 reverse | Eurofins | AAGACAGGAGAGCTGCAACTC |
| IFNL2/3 forward | Eurofins | GCCACATAGCCCAGTTCAAG |
| IFNL2/3 reverse | Eurofins | TGGGAGAGGATATGGTGCAG |
| CoV2 forward | Eurofins | GCCTCTTCTCGTTCC |
| Cov2 reverse | Eurofins | AGCAGCATCACCGCC |
| ISG15 forward | Eurofins | CCTCTGAGCATCCTGGT |
| ISG15 reverse | Eurofins | AGGCCGTACTCCCCCAG |
| IFIT1 forward | Eurofins | AAAAGCCCACATTTGAGGTG |
| IFIT1 reverse | Eurofins | GAAATTCCTGAAACCGACCA |
| OLMF4 forward | Eurofins | ACCTTTCCCGTGGACAGAGT |
| OLMF4 reverse | Eurofins | TGGACATATTCCCTCACTTTGGA |
| MUC-2 forward | Eurofins | TGTAGGCATCGCTCTTCTCA |
| MUC-2 reverse | Eurofins | GACACCATCTACCTCACCCG |
| CYP34A forward | Eurofins | GATGGCTCTCATCCCAGACTT |
| CYP3A4 reverse | Eurofins | AGTCCATGTGAATGGGTTCC |
| SI forward | Eurofins | AATCCTTTTGGCATCCAGATT |
| SI reverse | Eurofins | GCAGCCAAGAATCCCAAT |
| RNAscope® Probe - V-nCoV2019-S | ACD Bio | 848561 |
| RNAscope® Probe - Hs-ACE2 | ACD Bio | 848151 |
| RNAscope® Probe - Hs-ISG15 | ACD Bio | 311021 |
| Illumina_reverse_primer_SI-GA-H12_1 | Eurofins | CAAGCAGAAGACGGCATACGAGATGACAG<br>CATGTCTCGTGGGCTCGG |
| Illumina_reverse_primer_SI-GA-H12_2 | Eurofins | CAAGCAGAAGACGGCATACGAGATTTTGT<br>ACAGTCTCGTGGGCTCGG |
| Illumina_reverse_primer_SI-GA-H12_3 | Eurofins | CAAGCAGAAGACGGCATACGAGATAGGCC<br>GTGGTCTCGTGGGCTCGG |
| Illumina_reverse_primer_SI-GA-H12_4 | Eurofins | CAAGCAGAAGACGGCATACGAGATCCATA<br>TGCGTCTCGTGGGCTCGG |
| Illumina_reverse_primer_SI-GA-H11_1 | Eurofins | CAAGCAGAAGACGGCATACGAGATGGCGA<br>GTAGTCTCGTGGGCTCGG |
| COVID19_outer | Eurofins | ACCACACAAGGCAGATGGGC |
| COVID19_inner | Eurofins | GTCTCGTGGGCTCGGAGATGTGTATAAGA<br>GACAGCATTTTCACCGAGGCCACGC |
| ACE2_inner | Eurofins | GTCTCGTGGGCTCGGAGATGTGTATAAGA<br>GACAGTCCAGGGAACAGGTAGAGGAC |
| ACE2_outer | Eurofins | CTGGGAACTGGTGTAGCTGCA |

**Reagents and Tools table** (continued)

| Reagent/Resource | Reference or Source | Identifier or Catalog Number |
|---|---|---|
| APOA4_inner | Eurofins | GTCTCGTGGGCTCGGAGATGTGTATAAGA GACAGGGCCACTTGAGCTTCCTGGAG |
| APOA4_outer | Eurofins | CGAGGGGCTGCAGAAGTCAC |
| CHGB_inner | Eurofins | GTCTCGTGGGCTCGGAGATGTGTATAAGA GACAGCTGTCATTGGAGCGGTGGGC |
| CHGB_outer | Eurofins | TCTGAGGAGCCGGTGAGCAC |
| FABP6_inner | Eurofins | GTCTCGTGGGCTCGGAGATGTGTATAAGA GACAGTGGTCCCAGCACTACTCCGG |
| FABP6_outer | Eurofins | AGCATGGCTTTCACCGGCAA |
| FCGBP_inner | Eurofins | GTCTCGTGGGCTCGGAGATGTGTATAAGA GACAGCTCCCTGCTAGTCCGCCAGA |
| FCGBP_outer | Eurofins | CCTGGTACCGTGTAGTTGCCG |
| ISG15_inner | Eurofins | GTCTCGTGGGCTCGGAGATGTGTATAAGA GACAGGCTGGTGGTGGACAAATGCG |
| ISG15_outer | Eurofins | GCAGCTCCATGTCGGTGTCA |
| LGR5_inner | Eurofins | GTCTCGTGGGCTCGGAGATGTGTATAAGA GACAGCATCCAAACCAAATGGCTGTTAGGT |
| LGR5_outer | Eurofins | GCTGTGTTCTCTCTGGATAACCCA |
| LYZ_inner | Eurofins | GTCTCGTGGGCTCGGAGATGTGTATAAGA GACAGGGCAAAATACCAGCTGATGAAGGC |
| LYZ_outer | Eurofins | GCCCGGCCACATTCAGTTCT |
| MKI67_inner | Eurofins | GTCTCGTGGGCTCGGAGATGTGTATAAGA GACAGTCCACTTTGCCCCCTGTCCT |
| MKI67_outer | Eurofins | GCTCTGCTCCCGCCTGTTTT |
| OLFM4_inner | Eurofins | GTCTCGTGGGCTCGGAGATGTGTATAAGA GACAGTGCCTTTGTTTAAGCCTGGAACT |
| OLFM4_outer | Eurofins | ACCAACCCCCTTCTACTGCCT |
| SLC2A2_inner | Eurofins | GTCTCGTGGGCTCGGAGATGTGTATAAGA GACAGTCACCTGATCATATAGCGTGGGT |
| SLC2A2_outer | Eurofins | TGGCTAGTGGCAATAAGTTCCA |
| SMOC2_inner | Eurofins | GTCTCGTGGGCTCGGAGATGTGTATAAGA GACAGGGCCCCAAATCCCCTGAGAC |
| SMOC2_outer | Eurofins | TTCATCCGAGTGGCACTGGC |
| SST_inner | Eurofins | GTCTCGTGGGCTCGGAGATGTGTATAAGA GACAGAGCTGCTGTCTGAACCCAACC |
| SST_outer | Eurofins | CTGCTGATCCGCGCCTAGAG |
| **Chemicals, Enzymes and other reagents** | | |
| Advanced DMEM/F12 | Thermo Fischer Scientific | Cat# 12634010 |
| HEPES | Thermo Fischer Scientific | Cat3 15630080 |
| Penicillin/Streptomycin | Thermo Fischer Scientific | Cat#15140122 |
| GlutaMAX | Thermo Fischer Scientific | Cat# 35050061 |
| EDTA | Sigma Aldrich | Car#E9884 |
| MatriGel. GFR, LDEV free | Corning | Cat#354230 |
| B27 | Thermo Fischer Scientific | Cat#17504-044 |
| N-acetyl-cysteine | Sigma Aldrich | Cat# A9165 |
| Recombinant mouse EGF | Thermo Fischer Scientific | Cat# PMG8043 |
| [Leu15]-Gastrin I | Sigma-Aldrich | Cat# G9145 |
| A83-01 | Tocris | Cat#2939 |
| Recombinant human IGF-1 | BioLegend | Cat#590904 |

 

**Reagents and Tools table**   (continued)

| Reagent/Resource | Reference or Source | Identifier or Catalog Number |
|---|---|---|
| Recombinant human FGF basic | Peprotech | Cat#100-18B |
| Y-27632 | Caymann Chemicals | Cat#10005583 |
| Mouse recombinant noggin | Peprotech | Cat#250-38 |
| Collagen from human placenta | Sigma Aldrich | Cat#C5533-5MG |
| 0.05% Trypsin-EDTA | Thermo Fischer Scientific | Cat#25300054 |
| iTaq Universal SYBR green Supermix | BioRad | Cat#1725120 |
| Parafolmaldehyde | Sigma Aldrich | Cat#158127 |
| Triton X-100 | Sigma Aldrich | Cat#X100 |
| DAPI | Sigma Aldrich | Cat#D9542 |
| Fetal Bovine Serum | Capricorn | Cat#FBS-11A |
| DMEM, high glucose | Thermo Fischer Scientific | Cat#11965092 |
| DMEM/F12 | Thermo Fischer Scientific | Cat#11320033 |
| Draq5 | Abcam | Cat#ab108410 |
| Human recombinant IFN-beta 1 | Biomol | Cat#86421 |
| TrypLE Express | Thermo Fischer Scientific | Cat#12605036 |
| **Software** | | |
| Seurat R package v3.4.0 | https://satijalab.org/seurat/<br>Stuart *et al* (2019) | |
| pySCENIC v0.10.13 | https://github.com/aertslab/pySCENIC<br>Van de Sande *et al* (2020) | |
| destiny R package v3.4.0 | https://bioconductor.org/packages/release/bioc/<br>html/destiny.html<br>Angerer *et al* (2016) | |
| RNAscope HiPlex Image Registration | ACD Bio | |
| Fiji | https://imagej.net/Fiji<br>Schindelin *et al* (2012) | |
| Ilastik v1.3.3 | https://www.ilastik.org/download.html<br>Berg *et al* (2019) | |
| CellProfiler v4.1.3 | https://cellprofiler.org/releases<br>McQuin *et al* (2018) | |
| CellRanger v3.1.0 | https://support.10xgenomics.com/single-cell-ge<br>ne-expression/software/pipelines/latest/installa<br>tion | |
| TAPseq v1.2.0 | http://bioconductor.org/packages/release/bioc/<br>html/TAPseq.html | |
| R v3.6.2 | https://cran.r-project.org/ | |
| sctransform v0.3.2 | https://github.com/ChristophH/sctransform | |
| Drop-seq tools v1.13 | http://mccarrolllab.org/dropseq/ | |
| STAR v2.5.3a | https://github.com/alexdobin/STAR | |
| MAST v1.16 | https://github.com/RGLab/MAST<br>Finak *et al* (2015) | |
| EnrichR | https://maayanlab.cloud/Enrichr/ | |
| PROGENy v1.12 | https://bioconductor.org/packages/3.12/bioc/<br>html/progeny.html | |
| SoupX v1.5.0 | https://github.com/constantAmateur/SoupX | |
| **Other** | | |
| RNAeasy RNA extraction kit | Qiagen | Cat#74104 |
| iSCRIPT cDNA synthesis kit | BioRad | Cat#1708890 |

**Reagents and Tools table**   (continued)

| Reagent/Resource | Reference or Source | Identifier or Catalog Number |
| --- | --- | --- |
| Chromium Next GEM Chip G Single Cell Kit | 10X Genomics | 1000127 |
| Chromium Next GEM Single Cell 3′ Kit v3.1 | 10X Genomics | 1000268 |
| Chromium Controller & Next GEM Accessory Kit | 10X Genomics | 1000202 |
| KAPA Biosystems HiFi HotStart ReadyMix | Roche | KK2601 |
| SPRIselect | Beckman coulter | B23319 |

## Methods and Protocols

### Cells

T84 human colon carcinoma cells and their knockout derivative clones were maintained in a 50:50 mixture of Dulbecco's modified Eagle's medium (DMEM) and F12 (GibCo) supplemented with 10% fetal bovine serum and 1% penicillin/streptomycin (Gibco). Vero E6 and Caco-2 cells were maintained in DMEM supplemented with 10% fetal bovine serum and 1% penicillin/streptomycin. The mCherry-tagged Mx1 promoter plasmid was a kind gift from Ronald Dijkman (University of Bern), which was used to generate a T84 stable cell line via lentiviral transduction. Single clones were derived from this cell line and evaluated for their ability to respond to both type I and type III interferons.

Knockout of IRF3 in T84 cells was achieved by using the CRISPR/Cas9 system. Three different single-guide RNAs (sgRNAs) per gene were used targeting the coding region of IRF3 and inserted into the lentiviral vector lentiCRISPR v2 also encoding the Cas9 nuclease. Lentiviruses were produced, and T84 cells were transduced two times using 1:2 diluted stocks of lentiviral particles. Following puromycin selection, clonal selection was performed via single-cell dilution in a 96-well plate. Knockouts were confirmed by Western blot and functional assays.

### Viruses

SARS-CoV-2 (strain BavPat1) was obtained from the European Virology Archive. The virus was amplified in Vero E6 cells. Human astrovirus 1 was a kind gift from Stacy Schultz-Cherry (St. Jude) and was amplified in Caco-2 cells.

### Human organoid cultures and ethic approval

Human tissue was received from colon resection or ileum biopsies from the University Hospital Heidelberg. This study was carried out in accordance with the recommendations of the University Hospital Heidelberg with informed written consent from all subjects in accordance with the Declaration of Helsinki. All samples were received and maintained in an anonymized manner. The protocol was approved by the "Ethics commission of the University Hospital Heidelberg" under the protocol S-443/2017. Organoids were prepared following the original protocol described by (Sato *et al*, 2011). In brief, stem cells containing crypts were isolated following 2 mM EDTA dissociation of tissue samples for 1 h at 4°C. Crypts were spun and washed in ice cold PBS. Fractions enriched in crypts were filtered with 70 mM filters, and the fractions were observed under a light microscope. Fractions containing the highest number of crypts were pooled and spun again. The supernatant was removed, and crypts were resuspended in Matrigel. Crypts were passaged and maintained in basal and differentiation culture media as previously described (Stanifer *et al*, 2020c).

### 2D organoid seeding

1  8-well iBIDI glass-bottom chambers were coated with 250uL of 2.5% human collagen in water for 1 h prior to organoids seeding. Make sure that the well is fully coated to ensure even distribution of organoids.

2  Organoids were seeded at a rate to reach 60–70% confluency (around 70–80 organoids/well). The organoids should be large prior to seeding. If organoids are too small then they will not survive the digestion and re-seeding.

3  Collected organoids were spun at 450 xg for 5 min and the supernatant was removed. Careful not to take the pellet. The supernatant should be removed with a pipette and not a vacuum.

4  Organoids were washed 1X with cold PBS and spun at 450 xg for 5 min. PBS was removed and organoids were digested with 0.5% Trypsin-EDTA for 5 min at 37°C.

5  Digestion was stopped by addition of serum-containing medium.

6  Organoids were spun at 450 xg for 5 min, the supernatant was removed, and organoids were resuspended in normal growth media at a ratio of 250 µl media/well.

7  The collagen mixture was removed from the iBIDI chambers, and 250 µl of organoids was added to each well.

8  24 h post-seeding organoids should appear flat. There may be a lot of dead cells. If there are a lot of dead cells, media should be replaced.

9  48 h post-seeding, media was removed and replaced with differentiation medium.

10  Organoids were allowed to differentiate for four days prior to infection.

11  Differentiation was confirmed by qPCR for cell type markers (Appendix Fig S1A).

### Viral infections

Media was removed from cells and $10^6$ pfu of SARS-CoV-2 (as determined in Vero cells) was added to cells for 1 h at 37°C. Virus was removed, cells were washed 1x with PBS, and fresh media was added back to the cells. Virus infection was analyzed at time points indicated in the figure legends.

For astrovirus infection, media was removed from cells and $1.5\times10^5$ pfu astrovirus (as determined in T84 cells) in fresh media was added to the cells. Cells were incubated at 37°C and virus infection was analyzed 16 h post-infection.

### RNA isolation, cDNA, and qPCR

RNA was harvested from cells using RNAeasy RNA extraction kit (Qiagen) as per manufacturer's instructions. cDNA was made using iSCRIPT reverse transcriptase (Bio-Rad) from 250 ng of total RNA as

per manufacturer's instructions. qPCR was performed using iTaq SYBR green (Bio-Rad) as per manufacturer's instructions, TBP was used as a housekeeping gene. Primers used can be found in the tools table.

### Indirect immunofluorescence assay

At indicated times post-infection, cells were fixed in 4% paraformaldehyde (PFA) for 20 min at room temperature (RT). Cells were washed and permeabilized in 0.5% Triton X for 15 min at RT. Mouse monoclonal antibody against SARS-CoV NP and mouse monoclonal against J2 (scions) were diluted in phosphate-buffered saline (PBS) at 1/1000 dilution and incubated for 1 h at RT. Cells were washed in 1X PBS three times and incubated with secondary antibodies conjugated with AF488, or AF568 directed against the animal source) and DAPI for 45 min at RT. Cells were washed in 1X PBS three times and maintained in PBS. Cells were imaged by epifluorescence on a Nikon Eclipse Ti-S (Nikon).

### In-cell Western (TCID50 assay)

20,000 Vero E6 cells were seeded per well into a 96-well dish 24 h prior to infection. 100 μl of harvested supernatant was added to the first well. Seven serial 1:10 dilutions were made (all samples were performed in triplicate). Infections were allowed to proceed for 24 h. 24 h post-infection cells were fixed in 2% PFA for 20 min at RT. PFA was removed and cells were washed twice in 1X PBS and then permeabilized for 10 min at RT in 0.5% Triton X. Cells were blocked in a 1:2 dilution of Li-Cor blocking buffer (Li-Cor) for 30 min at RT. Cells were stained with 1/1000 dilution anti-dsRNA (J2) for 1 h at RT. Cells were washed three times with 0.1% Tween in PBS. Secondary antibody and DNA dye Draq5 were diluted 1/10,000 in blocking buffer and incubated for 1 h at RT. Cells were washed three times with 0.1% Tween/PBS. Cells were imaged in 1X PBS on a LICOR imager.

### Organoid dissociation for scRNAseq

2D seeded organoids harvested after 0 (mock), 12, and 24 h post-infection were washed in cold PBS and incubated in TrypLE Express for 25 min at 37°C. When microscopic examination revealed that cells had reached a single-cell state, they were resuspended in DMEM/F12 and spun at 500 xg for 5 min. Supernatant was removed and the cell pellet was resuspended in PBS supplemented with 0.04% BSA and passed through a 40 μm cell strainer. Resulting cell suspensions were used directly for single-cell RNA-seq.

### Single-cell RNA-seq library preparation

Single-cell suspensions were loaded onto the 10x Chromium controller using the 10x Genomics Single Cell 3' Library Kit NextGem V3.1 (10x Genomics) according to the manufacturer's instructions. In summary, cell and bead emulsions were generated, followed by reverse transcription, cDNA amplification (5 μl of amplified cDNA was set apart for targeted scRNAseq amplification), fragmentation, and ligation with adaptors followed by sample index PCR. Resulting libraries were quality checked by Qubit and Bioanalyzer, pooled, and sequenced using HiSeq4000 (Illumina; high-output mode, paired-end 26 x 75 bp).

### Targeted single-cell RNA sequencing

For targeted scRNAseq, outer and inner primers for targeted amplification were designed using an R package for primer design described

in (Schraivogel *et al*, 2020) and available through Bioconductor (http://bioconductor.org/packages/release/bioc/html/TAPseq.html). Primers were ordered desalted as ssDNA oligonucleotides and pooled in an equimolar amount, except for the primer targeting SARS-CoV-2 mRNA which was added in eightfold excess to the outer and inner panel. All primer sequences are described in Table EV1. Targeted scRNAseq was performed as previously described (Schraivogel *et al*, 2020), except for using amplified cDNA from the 10X Genomics 3' scRNAseq protocol as input material. In short, 10 ng of amplified cDNA were used as input for the outer primer PCR and amplified with 10 PCR cycles. A second semi-nested PCR using 10 ng of Ampure purified outer PCR as input was performed with inner primer mix and seven cycles of PCR. Then, a third PCR was done adding Illumina adapters. Resulting libraries were quality checked by Qubit and Bioanalyzer, pooled, and sequenced using HiSeq4000 (Illumina; high-output mode, paired-end 26 x 75 bp).

### Pre-processing and quality control of scRNAseq data

Raw sequencing data were processed using the CellRanger software (version 3.1.0). Reads were aligned to a custom reference genome created with the reference human genome (GRCh38) and SARS-CoV-2 reference genome (NC_045512.2). The resulting unique molecular identifier (UMI) count matrices were imported into R (version 3.6.2) and processed with the R package Seurat (version 3.1.4). Low-quality cells were removed, based on the following criteria that maintains the distribution in each sample. All cells with mitochondrial reads > 30% were excluded. Second, we limited the acceptable numbers of detected genes. For both types of samples, cells with < 1,500 or > 9,000 detected genes were discarded. The remaining data were further processed using Seurat. To account for differences in sequencing depth across cells, UMI counts were normalized and scaled using regularized negative binomial regression as part of the package sctransform. Afterward, ileum and colon organoids samples were integrated independently to minimize the batch and experimental variability effect. Integration was performed using the *IntegrateData* function from Seurat. The resulting SCT corrected counts were used only for UMAP visualization and clustering downstream analysis and the non-integrated counts for any quantitative comparison. Such as box plots.

### Pre-processing of targeted scRNAseq

Targeted scRNAseq pre-processing was done as described in (Schraivogel *et al*, 2020). In summary, following demultiplexing by sample, sequencing data were processed following the workflow provided by Drop-seq tools (v. 1.13, http://mccarrolllab.org/dropseq/) with STAR (v. 2.5.3a) to align reads. To mitigate potential multi-mapping issues, targeted samples were aligned to a custom alignment reference containing only genes of the respective target gene panel, including the SARS-CoV-2 genome. This reference contained the sequences of all target gene loci as individual contigs with overlapping loci merged into one contig. UMI observations were extracted using the Drop-seq tools GatherMolecularBarcodeDistributionByGene program. A custom script (Python v. 3.6.6) was used to filter for chimeric reads with a transcripts-per-transcript (TPT) cutoff of 0.25, and UMI observations were converted to transcript counts. Cell-containing droplets were extracted using the filtered cell barcodes from the scRNASeq data. Other detected cell barcodes droplets were categorized as empty droplets. The infection status for

every cell was extracted from the targeted gene expression data by thresholding the SARS-CoV-2 counts using the media expression of the cell containing droplets (Appendix Fig S2). Additionally, SoupX v.3.1.5 (Young & Behjati, 2020) was evaluated as a tool for correcting viral contamination. Furthermore, Pearson correlation of each targeted gene to its WTA equivalent was calculated.

### Clustering and identification of cell type markers

We performed principal component analysis (PCA) using 3,000 highly variable genes (based on average expression and dispersion for each gene). The top 30 principal components were used to construct a shared nearest neighbor (SNN) graph and modularity-based clustering using the Louvain algorithm was performed to obtain 24 and 19 clusters in the colon and ileum organoids, respectively. Finally, Uniform manifold approximation and projection (UMAP) visualization was calculated using 30 neighboring points for the local approximation of the manifold structure. Marker genes for every cell type were identified by comparing the expression of each gene in a given cluster against the rest of the cells using the receiver operating characteristic (ROC) test. To evaluate which genes classify a cell type, the markers were selected as those with the highest classification power defined by the AUC (area under the ROC curve). These markers along with canonical markers for intestinal and colonic cells were used to annotate each of the clusters of the ileum and colon sample merging similar clusters into the major cell types they belonged to. The comparison between the shared cell types from the infected and mock organoids was performed by calculating the Pearson correlation between the averaging normalized gene expression of each cell type. Moreover, The Seurat label transfer routine was used to map the cell types from colon (Smillie et al, 2019) and ileum (preprint: Triana et al, 2020) tissue single-cell datasets to the respective organoid cells.

### Differential expression analysis

To identify the changes in expression across conditions. Differential expression tests were performed using MAST (Finak et al, 2015). To reduce the size of the inference problem and avoid cell proportion bias, separate models were fit for each cell lineage and comparisons between mock, 12 h, and 24 h post-infection were performed. False discovery rate (FDR) was calculated by the Benjamini–Hochberg method (Benjamini & Hochberg, 1995) and significant genes were set as those with FDR of less than 0.05. Subsequently, genes whose mRNAs were found to be differentially expressed were subjected to a gene set overrepresentation analysis using the EnrichR package in R. Furthermore signaling pathway enrichment was calculated using PROGENy.

### Multiplex FISH sample preparation

1. Organoids were seeded in expansion medium on glass coverslips. At 24 h post-seeding, the expansion medium was replaced by differentiation medium and organoids were left to differentiate for 4 days.
2. qPCR validation was conducted to ensure organoids were differentiated and contained all expected cell types, organoids were infected, harvested after 12 and 24 hpi, and fixed in 4% PFA for 30 min.
3. HiPlex (RNAscope) was performed following the manufacturer's instructions. Fixed samples were dehydrated sequentially with 50%, 70%, and 100% ethanol for 5 min each.

4. Samples were permeabilized with Protease III for 30 min. Once permeabilized be careful to not allow the sample to dry out for the remainder of the assay.
5. All the HiPlex probes were hybridized and amplified together using manufacturer specified HybEZ II Hybridization system. Probes were designed for genes identified as cell type markers and/or corroborated by literature.
6. After hybridization, samples were washed and counterstained with DAPI, mounted with ProLong Gold Antifade Mountant.
7. Imaging was performed with Plan Flour 20x objective mounted on the Nikon Ti-E inverted microscope (Nikon Instruments) in fluorescence (DAPI, GFP, Cy3, and Cy5) channels. The microscope was controlled using the Nikon NIS Elements software.
8. After each round of imaging, the mountant was removed by soaking in 4x SSC buffer for 30 min, fluorophores were cleaved and washed using manufacturer specified cleaving solution and wash buffer. Washed samples were prepared for the second round of hybridization and imaging, repeating for a total of four rounds.
9. All images from all rounds of staining were then registered with each other to generate images using HiPlex image registration software (ACD Bio). Further brightness and contrast adjustments were performed using Fiji.

### Multiplex FISH data analysis

1. The HiPlex probe fluorescent signal was used to determine the ACE2 and ISG15 RNA expression levels, as well as the SARS-CoV-2 infection levels.
2. To obtain a resolution at a single-cell level, first nuclei segmentation and classification was done on raw DAPI images using the Pixel Classification + Object Classification workflow from ilastik 1.2.0. Ilastik is based on machine learning algorithms where the user teaches the program what nuclei and what background is. Therefore, to obtain a precise segmentation, it is important to use at least 10 raw DAPI images that cover the different seeding conditions (e.g. very densely seeded cells to less dense).
3. The resulting Object Prediction masks represented all nuclei as individual objects in a 2D plane and were saved as 16bit Tagged Image File Format.
4. To measure the single-cell fluorescent intensity for the ACE2, ISG15, and SARS-CoV-2 probes, a pipeline using CellProfiler 3.1.9 was developed. Briefly, first the raw grayscale images corresponding to the ACE2, ISG15, and SARS-CoV-2 probe fluorescent signals were uploaded on the pipeline. These images were specified as images to be measured. The corresponding Object Prediction masks previously generated by ilastik were then uploaded, converted into binary nuclei masks and used to define the objects to be measured. Finally, with a MeasureObjectIntesity module the fluorescence intensity features, the cell number and the single-cell localization were measured for the identified objects from the binary nuclei mask.
5. The outcome was exported to a spreadsheet and contained the localization as well as the mean intensity units rescaled from 0 to 1 of ACE2, ISG15, and SARS-CoV-2 fluorescent signals for each single cell.
6. To determine the infection status for every cell, a threshold was calculated using the SARS-CoV-2 mean fluorescent intensity signal of mock-treated versus representative infected cells.

The threshold was set to 0.015 mean intensity units. When setting the threshold, it is important to control that the percentage of SARS-CoV-2-positive cells obtained from the analysis fits the raw imaging data. To this end, random images were picked; positive cells were counted and compared with the analysis results.

7    In next step, the ACE2 signal was further processed. Due to the probe quality, the ACE2 fluorescent signal showed strong variations between different images and hence technical replicates. To minimize the variability, for each individual image the ACE2 mean intensity signal was normalized and rescaled from 0 to 1, 0 corresponding to the lowest and 1 to the highest ACE2 mean intensity signal of a cell from the corresponding image.

8    Finally, the SARS-CoV2 mean intensity signal was plotted against the normalized ACE mean intensity signal or the ISG15 mean intensity signal using GraphPad Prism version 6.0.

### Transcription factor activity along diffusion map pseudotime

Raw counts were normalized using the SCTransform method implemented in Seurat v. 3.3.4, regressed over UMI counts. Transcription factor activities were then calculated using pySCENIC v 0.10.13 (Van de Sande *et al*, 2020). Independently, the diffusion maps were computed using the destiny R library v 3.2.0. For the inference of pseudotime as input, we used a set of curated genes related to IFN signaling from Reactome (R-HSA-913531). Visualization of the TF activities along trajectories was carried out with custom R scripts (v 4.0.2).

# Data availability

The raw sequencing and count matrices generated during this study are available at the NCBI Gene Expression Omnibus (accession no. GSE156760). https://www.ncbi.nlm.nih.gov/geo/query/acc.cgi?acc=GSE156760).

Datasets including raw and integrated gene expression data, cell type annotation, metadata, and dimensionality reduction are available as Seurat v3 objects through figshare. (https://doi.org/10.6084/m9.figshare.13703752.v1).

Expanded View for this article is available online.

### Acknowledgements

This work was supported by research grants from the Deutsche Forschungsgemeinschaft (DFG): project numbers 415089553 (Heisenberg program), 240245660 (SFB1129), 278001972 (TRR186), and 272983813 (TRR179), The State of Baden Wuerttemberg (AZ: 33.7533.-6-21/5/1) and the Bundesministerium Bildung und Forschung (BMBF) (01KI20198A) to SB. MS was supported by the DFG (416072091) and the BMBF (01KI20239B). CMZ is supported by the SFB1129 (240245660), CR and CH are supported by the TRR179 (272983813). We also acknowledge funding from the Helmholtz International Graduate School for Cancer Research to CK, the German Academic Exchange Service (DAAD) (Research Grant 57440921) to PD, Darwin Trust of Edinburgh to ST, and the ERC Consolidator Grant METACELL (773089) to TA. Open Access funding enabled and organized by Projekt DEAL.

### Author contributions

Experiments, single-cell data analysis, and manuscript writing: ST; Single-cell data analysis: CR and AS; Experiments: CMZ, CK, PD, MS, and MLS; Target scRNA0seq: DS and ARG; Concept of the study and critical discussions: LS and CH; Experiment conception, result interpretation, and manuscript writing: SB, MLS, and TA. All authors: Final approval of the manuscript.

### Conflict of interest

The authors declare that they have no conflict of interest.

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
