## [Review Process File · Molecular Systems Biology]

Single-cell analyses reveal SARS-CoV-2 interference with intrinsic immune response in the human gut

Sergio Triana, Camila Metz-Zumaran, Carlos Ramirez, Carmon Kee, Patricio Doldan, Mohammed Shahrzaz, Daniel Schraivogel, Andreas Gschwind, Ashwini Kumar Sharma, Lars Steinmetz, Carl Herrmann, Theodore Alexandrov, Steeve Boulant, and Megan Stanifer

DOI: 10.15252/msb.202110232

Corresponding author(s): Megan Stanifer (m.stanifer@dkfz.de), Theodore Alexandrov (theodore.alexandrov@embl.de), Megan Stanifer (m.stanifer@dkfz.de)

Review Timeline:

Submission Date:	19th Jan 21
Editorial Decision:	20th Jan 21
Revision Received:	9th Feb 21
Editorial Decision:	9th Mar 21
Revision Received:	21st Mar 21
Editorial Decision:	29th Mar 21
Revision Received:	31st Mar 21
Accepted:	31st Mar 21

Editor: Maria Polychronidou

Transaction Report:

The reviewers' comments and authors' responses are not available with this article, as the initial review process took place with another journal.

Thank you again for sending us your manuscript "Single-cell analyses reveal SARS-CoV-2 interference with intrinsic immune response in the human gut" and the reviews you received at the other journal. I think that the reviews seem thorough and constructive and we can use them rather than reviewing the study from scratch. In line with the comments of the reviewers, I think that the scRNA-seq analyses are relevant for exploring the response to SARS-CoV-2 in the intestine. We would be happy to consider the study and I would therefore invite you to submit it once you have completed the revisions addressing the issues raised. Please include a detailed point by point response, so that we can easily evaluate the changes.

In summary, the main issues raised were mostly technical and refer to the need to:

- provide further evidence that the used organoids have properly differentiated
- better support that the infected vs uninfected cells have been correctly determined
- clarify/better support that annotation and clustering processes
- better support the conclusions regarding transcriptional repression

We think that it would be important to address the technical issues delineated above. Given the technical nature of the concerns I think that we would need to consult an external expert (e.g. by a member of our Editorial Advisory Board or an expert in the field) and ask them to look at the performed revisions.

Specific answers to the editor:

1) provide further evidence that the used organoids have properly differentiated

We have now added a panel (Fig. S1A) to show that our two dimensional organoids fully differentiate and have clarified this in the methods section. We have also added references that also use a two dimensional approach with differentiated organoids for infectious disease research.

2) better support that the infected vs uninfected cells have been correctly determined

We have now better justified why our approach of removing a baseline of virus counts in the Targeted scRNAseq allows us to correctly determine the infected cells. In response to the reviewer suggestion, we evaluated SoupX and showed this approach is not suitable to correct the detection of contaminating viral transcript in our datasets, as this falsely enriches small cells (please see specific answer to reviewers).

3) clarify/better support that annotation and clustering processes

We added figure S3, here we clarify the different steps we used for the clustering and annotation of our colon and ileum organoids and we showed the top markers differentially expressed in each cluster. Importantly, we also show the correlation between the cell types from mock sample and infected sample and the association between our annotations and the previously reported annotation in the equivalent tissue cell atlas. Finally, with respect to the presence of immature enterocyte 2, we now better refer to our previous work where we have clearly identified immature enterocyte2 in human ileum biopsy, further suggesting that we have properly annotated our data. There is currently a lack of consensus in the field with respect to the precise identity and sub-clusters of cells expected to be found in intestinal organoids. This is mostly due to the fact that there just a handful of reports describing scRNAseq of human intestinal organoids. To further justify our clustering we would like to direct you on the resources website of Alex Shalek lab (world leading expert in scRNAseq at MIT) where one could find a UMAP of intestinal organoids clearly showing immature enterocytes (the shalek lab refer to them as early enterocyte).
<http://shaleklab.com/resource/covid-19-resources/>

4) better support the conclusions regarding transcriptional repression

We have clarified in the text that our experiments reporting inhibition of IFN-mediated signaling using different multiplicity of infection (Figure 6) were analyzed by normalizing to a housekeeping gene (which will then compensate in case of global transcription repression). Most importantly, we have added the Fig.S2 and S5C showing that the modulation of markers we observed is not a consequence of a global down-regulation of gene expression modulated by the virus, as can be seen by the similar number of detected transcripts of treated and untreated cells.

****The authors' responses to the reviewers' comments are not available with this article, as the initial review process took place with another journal.***

Thank you again for submitting your work to Molecular Systems Biology. We have now heard back from the reviewer who was asked to evaluate your study. As we previously discussed, the reviewer was given access to the manuscript and your responses to the reviewers' comments from the other journal. They were asked to evaluate whether the reviewers' concerns have been adequately addressed, and to assess the suitability of the revised study for publication keeping in mind the editorial criteria of Molecular Systems Biology. As you will see below, the reviewer is overall supportive. However, they recommend some modifications, which we would ask you to perform in a revision.

Most issues raised refer to the need to include further discussions and clarifications in order to present the potential limitations of the study in a more balanced way, given that it is performed in organoids and not in patients. Regarding the identification of infected vs bystander cells, an issue that was prominent in the previous reports, the reviewer strongly recommends including supporting data using an orthogonal method, to better support the related conclusions.

On a more editorial level, we would ask you to address the following.

REFEREE REPORTS

Reviewer #1:

In this manuscript, the authors study the intestinal tropism of SARS-CoV-2 by following the infection in a 2D model of primary human intestinal organoids. They studied the relationship between the levels of Ace2 expression with susceptibility and found a lack of correlation between these features. They further describe differences in induced proinflammatory gene expression signatures between infected and bystander cells.

I appreciated the discussion of the inappropriate usage of ACE2 expression as the only basis for conjectures about the infectability of cell types or organs. While Ace2 expression is easily measurable by scRNAseq, many virologists (Vincent Racaniello amongst others) have warned that a receptor is necessary but not sufficient for infection. The internal cellular environment has to play

a permissive role and further external factors are involved.

Several groups have used intestinal organoids to study SARS-CoV2 infection in the intestinal epithelium. While I certainly appreciate the feasibility of this approach, I'm not fully convinced of its relevance to COVID19 pathophysiology in humans. The weaknesses of the model and its application in this study have been raised by the previous round of reviews and has been partially addressed by the authors. I understand that studying the intestinal response to SARS-CoV2 at single-cell resolution in a more relevant context would be logistically challenging due to timing and biosafety issues and lies beyond the scope of the current manuscript. I am not opposed to the publication of this manuscript in MSB but I strongly recommend a discussion of the weaknesses and limitations of the used model system in the discussion section and discourage uncritical extrapolation of the findings to human disease.

Specific comments:

The proportions of cell types and states in intestinal organoids do not fully reflect the tissue of origin. Growth in the Sato medium results in strongly stem-cell biased conditions, the here employed differentiation protocol seems to induce more mature cells but (as mentioned by the previous reviewer 1) the lack of goblet cell capture in the colon organoids is worrying. The authors argue that their scRNAseq dataset of human intestinal organoids is consistent with the (limited) existing literature, yet this does not mean that organoids are an appropriate model for the study of cell tropism and will faithfully reflect the tissue of origin. This should be discussed.

Along similar lines: I am not convinced by the use of Cyp3a4 expression as a single qPCR target to justify "full" differentiation of the cultures. In their previous Cell Reports study the authors have used SI as marker for differentiation, why do they use Cyp3a4 here?

There is a continuum of gene expression along the Crypt-Villus trajectory in ileal enterocytes (or crypt bottom to top in colonocytes, respectively), either the authors should validate the expression of true top markers or they should mention in the discussion the remaining uncertainties regarding the in vivo relevance of the present results (since organoids are likely not reflecting all enterocyte states).

The identification of infected vs bystander cells is of importance in this manuscript. Both previous reviewers have questioned the applied approach with a hard threshold. I am not fully convinced by the novel explanations (I assume the authors referred to Figure S4 and not S3 as mentioned in their response to the reviewers). I understand the problems with the proposed SoupX approach, the resulting data should be presented in the supplementary figure. To clarify the issue and to increase confidence in the applied method I suggest an orthogonal measurement of the number of infected cells by quantifying fluorescently labeled infected organoids (as in Fig 3f) and use scRNAseq on the second half of the same sample for comparison.

Figure 7: the depiction of a human gut implies in vivo relevance of the presented findings. I have not seen evidence that the proposed interaction between infected and bystander cells is relevant outside of organoid cultures. I have also missed in vivo validation of the existence/relevance of the described organoid immature enterocyte states 1 and 2, so I ask the authors to replace the gut depiction with an organoid scheme and to mention this in the discussion.

Fig S3: heatmaps in b and g appear smoothed, this is a Matlab export / PDF bug, please correct.

For all following the comments are in **blue** and our responses are in **black**.

Response to editor comments:

Most issues raised refer to the need to include further discussions and clarifications in order to present the potential limitations of the study in a more balanced way, given that it is performed in organoids and not in patients. Regarding the identification of infected vs bystander cells, an issue that was prominent in the previous reports, the reviewer strongly recommends including supporting data using an orthogonal method, to better support the related conclusions.

We have now modified our discussion to avoid over interpretation of our organoid model and highlighted its limitations. The orthogonal approach (immunostaining) was already performed in the original version but we did not include a quantification of the number of infected cells. In the revised version, we have included this quantification (Fig EV1 and EV3) and are now commenting in the text that the numbers of infected cells are matching between the single cell experiments and our immunofluorescence. Importantly we have also highlighted that the immunostaining samples were performed in parallel to the sequencing.

Response to the reviewer comments:

We thank the reviewer for their comments and suggestions. We believe this new version is now strengthened, more clear and also less speculative in our conclusion to the implication of our findings to the “real” human gut. In short, we have addressed the main concern of the reviewer and have updated our text to show the

limitations of the organoids system compared to “real” primary human intestinal tissue. While we would have loved to do this study with primary patient material, as the reviewer mentioned this was out of the scope of this project. We have actively tried to assess biopsies of infected patients through our gastroenterologist collaborator. While a great number of patients had mild to severe diarrhea, we concluded that their gastroenteric symptoms did not justify the risks associated with the invasive process of taking a biopsy as many of the patients were on blood thinners and were at a high risk for these procedures.

Reviewer #1:

In this manuscript, the authors study the intestinal tropism of SARS-CoV-2 by following the infection in a 2D model of primary human intestinal organoids. They studied the relationship between the levels of Ace2 expression with susceptibility and found a lack of correlation between these features. They further describe differences in induced proinflammatory gene expression signatures between infected and bystander cells.

I appreciated the discussion of the inappropriate usage of ACE2 expression as the only basis for conjectures about the infectability of cell types or organs. While Ace2 expression is easily measurable by scRNAseq, many virologists (Vincent Racaniello amongst others) have warned that a receptor is necessary but not sufficient for infection. The internal cellular environment has to play a permissive role and further external factors are involved.

Several groups have used intestinal organoids to study SARS-CoV2 infection in the intestinal epithelium. While I certainly appreciate the feasibility of this approach, I'm not fully convinced of its relevance to COVID19 pathophysiology in humans. The weaknesses of the model and its application in this study have been raised by the previous round of reviews and has been partially addressed by the authors. I understand that studying the intestinal response to SARS-CoV2 at single-cell resolution in a more relevant context would be logistically challenging due to timing and biosafety issues and lies beyond the scope of the current manuscript. I am not opposed to the publication of this manuscript in MSB but I strongly recommend a discussion of the weaknesses and limitations of the used model system in the discussion section and discourage uncritical extrapolation of the findings to human disease.

We thank the reviewer for their comments. We agree that these models are important as they allow us to address cell lineage specific differences but they do have their limitations. The discussion has now been updated to reflect these limitations.

Specific comments:

The proportions of cell types and states in intestinal organoids do not fully reflect the tissue of origin. Growth in the Sato medium results in strongly stem-cell biased conditions, the here employed differentiation protocol seems to induce more mature cells but (as mentioned by the previous reviewer 1) the lack of goblet cell capture in the colon organoids is worrying. The authors argue that their scRNAseq dataset of human intestinal organoids is consistent with the (limited) existing literature, yet this does not mean that organoids are an appropriate model for the study of cell tropism and will faithfully reflect the tissue of origin. This should be discussed.

We agree with the reviewer that organoids do not fully recapitulate the tissue and are limited to the differentiation media that they are exposed. Unlike in the body, organoids are kept in a high stem cell state which allows them to constantly proliferate. The Sato differentiation media that we use induces differentiation but the ratio of the absorptive vs secretory lineages is different from in vivo. This has been now discussed in the text.

Along similar lines: I am not convinced by the use of Cyp3a4 expression as a single qPCR target to justify "full" differentiation of the cultures. In their previous Cell Reports study the authors have used SI as a marker for differentiation, why do they use Cyp3a4 here?

There is a continuum of gene expression along the Crypt-Villus trajectory in ileal enterocytes (or crypt bottom to top in colonocytes, respectively), either the authors should validate the expression of true top markers or they should mention in the discussion the remaining uncertainties regarding the in vivo relevance of the present results (since organoids are likely not reflecting all enterocyte states).

We have used several markers for the validation but only included a few. We have not updated the figure to include a few more makers (including SI) and have updated the discussion regarding the remaining uncertainties and limitations associated with organoids.

The identification of infected vs bystander cells is of importance in this manuscript. Both previous reviewers have questioned the applied approach with a hard threshold. I am not fully convinced by the novel explanations (I assume the authors referred to Figure S4 and not S3 as mentioned in their response to the reviewers). I understand the problems with the proposed SoupX approach, the resulting data should be presented in the supplementary figure. To clarify the issue and to increase confidence in the applied method I suggest an orthogonal measurement of the number of infected cells by quantifying fluorescently labeled infected organoids (as in Fig 3f) and use scRNAseq on the second half of the same sample for comparison.

We thank the reviewer for their comment as this was apparently unclear from our text. We have now updated the text to more clearly explain that all single cell experiments were also done in parallel with immunofluorescence and qPCR validation. Analyses of our single cell sequencing experiments revealed a similar number of infected cells compared to the direct determination of the infectivity by immunofluorescence as depicted in the time course of infection presented in Figure EV1. We have realized that the quantification of the number of infected cells from our infected organoids was missing in our original submission. We have now added a panel in EV1 showing the percentage of infected cells and the percent of infected cells was also added to the single cell data in EV3. Additionally, we have also added the SoupX data and made it into a new supplementary figure (Appendix Figure 2).

Figure 7: the depiction of a human gut implies *in vivo* relevance of the presented findings. I have not seen evidence that the proposed interaction between infected and bystander cells is relevant outside of organoid cultures. I have also missed *in vivo* validation of the existence/relevance of the described organoid immature enterocyte states 1 and 2, so I ask the authors to replace the gut depiction with an organoid scheme and to mention this in the discussion.

We thank the reviewer for this comment. It is true that organoids are not *in vivo* models and more work would need to be done to make precise statements about how infection impacts the gastrointestinal tract of COVID-19 patients. We have updated the figure and the text and toned down our statements in the discussion.

Fig S3: heatmaps in b and g appear smoothed, this is a Matlab export / PDF bug, please correct.

We have checked the image and from our side the heatmaps appear ok. It seems that maybe something happened in the compression of the file when sent to reviewers. Let us know if this problem occurs again in this version and we will upload these individual figures separately to the MSB website.

We would like to thank the reviewer once more for their time. We do appreciate all comments as we feel that the manuscript has now been improved in readability, has a more balanced conclusion avoiding overstatement of our findings to the human gut and we do feel that discussing the limitation of organoids is also important for the field.

Looking forward to hearing from you
Megan Stanifer and Steeve Boulant, on behalf of all co-authors.

Thank you for sending us your revised manuscript. We think that the performed revisions have satisfactorily addressed the issues raised by the reviewer. As such, we are glad to inform you that your study can soon be accepted for publication, pending some minor modifications listed below.

3rd Authors' Response to Reviewers**31st Mar 2021**

The authors have made all requested editorial changes.

Accepted**31st Mar 2021**

Thank you again for sending us your revised manuscript and for performing the last requested editorial changes. We are now satisfied with the modifications made and I am pleased to inform you that your paper has been accepted for publication.

Corresponding Author Name: Megan Stanifer, Steeve Boulant, Theodore Alexandrov

Journal Submitted to: MSB

Manuscript Number: MSB-2021-10232